# Contractile ring mechanosensation and its anillin-dependent tuning during early embryogenesis

Christina Rou Hsu[1,2,3], Gaganpreet Sangha [1,2,3], Wayne Fan[1,2], Joey Zheng[1,2] & Kenji Sugioka [1,2] ✉

Cytokinesis plays crucial roles in morphogenesis. Previous studies have examined how tissue mechanics influences the position and closure direction of the contractile ring. However, the mechanisms by which the ring senses tissue mechanics remain largely elusive. Here, we show the mechanism of contractile ring mechanosensation and its tuning during asymmetric ring closure of *Caenorhabditis elegans* embryos. Integrative analysis of ring closure and cell cortex dynamics revealed that mechanical suppression of the ring-directed cortical flow is associated with asymmetric ring closure. Consistently, artificial obstruction of ring-directed cortical flow induces asymmetric ring closure in otherwise symmetrically dividing cells. Anillin is vital for mechanosensation. Our genetic analysis suggests that the positive feedback loop among ring-directed cortical flow, myosin enrichment, and ring constriction constitutes a mechanosensitive pathway driving asymmetric ring closure. These findings and developed tools should advance the 4D mechanobiology of cytokinesis in more complex tissues.

The cytokinetic contractile ring physically partitions the dividing cell during cell division, but it also plays pivotal roles in morphogenesis by regulating its position and function along the body axis[1–3]. This is evident in processes such as asymmetric cell division and epithelial morphogenesis, where the position and closure of the ring are asymmetrically regulated to control the size, shape, and arrangements of daughter cells. Although previous studies have suggested that both intracellular and extracellular mechanics influence this "morphogenetic cytokinesis[4–6]", the mechanism by which the contractile ring senses mechanical cues and modulates its function in response remain largely unexplored.

One of the promising model systems for studying contractile ring mechanosensation is the asymmetric ring closure observed in animal zygotes and epithelial tissues, termed unilateral or asymmetric cytokinesis[7,8]. Unilateral cytokinesis is ubiquitously observed in both invertebrates and vertebrates, including humans[9–13], and its

dysregulation has been associated with reduced cytokinesis resilience, disrupted epithelial integrity, and defective lumen morphogenesis[9,14,15].

In some cases, unilateral cytokinesis occurs without mechanical regulation but rather through the non-homogenous activities of the RhoA signaling pathway at the cell cortex. The RhoA signaling pathway plays a central role in eukaryotic cytokinesis and begins with the activation of the small GTPase RhoA (Fig. 1a)[16,17]. Once activated, RhoA induces Rho-associated kinase (ROCK)-dependent myosin activation and formin-dependent actin assembly, leading to the formation of the contractile ring[16,18,19]. The mitotic spindle plays a critical role in specifying the site of RhoA activation. It promotes RhoA activation at the cell equator through the centralspindlin complex[20–23] but inhibits ring assembly in the polar region[24–31]. Consequently, the off-centering of the spindle along the transverse axis relative to the pole-to-pole axis induces furrowing at one side of the cell cortex (hereafter we call this site as the leading edge)[32–34].

[1]Life Sciences Institute, The University of British Columbia, 2350 Health Sciences Mall, Vancouver, BC V6T1Z3, Canada. [2]Department of Zoology, The University of British Columbia, 2350 Health Sciences Mall, Vancouver, BC V6T1Z3, Canada. [3]These authors contributed equally: Christina Rou Hsu, Gaganpreet Sangha. ✉e-mail: kenji.sugioka@ubc.ca

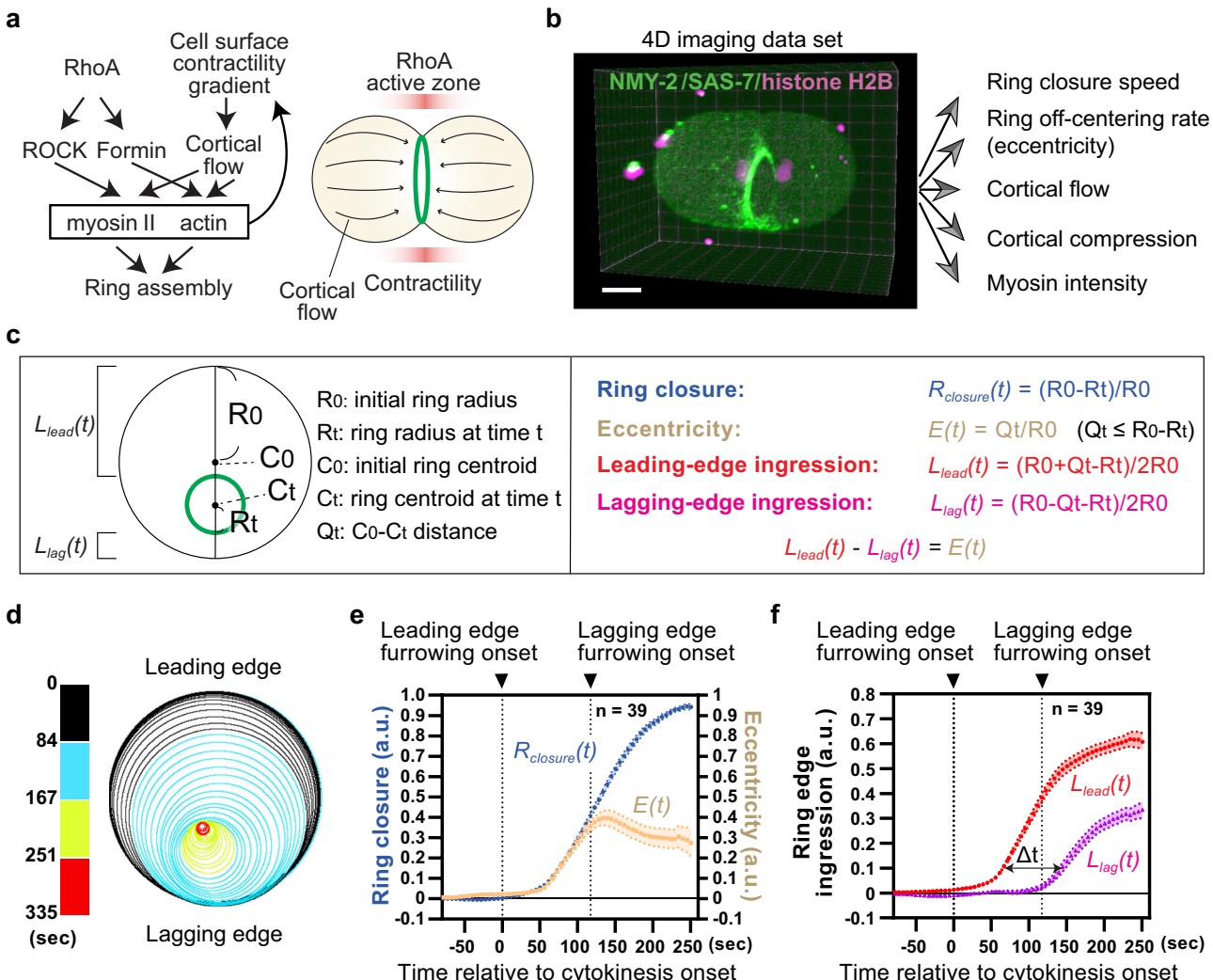

**Fig. 1 | High-resolution 4D analysis of asymmetric contractile ring closure in the *C. elegans* zygote. a** RhoA and cortical flow pathways that control contractile ring assembly. RhoA activation at the equatorial cell cortex stimulates downstream myosin activity and actin polymerization. A gradient in cell surface contractility induces cortical flow towards the ring, leading to myosin II enrichment and actin alignment. **b** Integrative 4D analysis of cortex-ring dynamics during asymmetric contractile ring closure. Using identical 4D imaging datasets of zygotes expressing NMY-2::GFP (green, non-muscle myosin II), GFP::SAS-7 (green, centriole) and histone H2B::mCherry (magenta, chromosome), multiple parameters were measured to understand the relationship between the contractile ring and cortical dynamics. In this and subsequent figures, anterior and posterior are to the left and right, respectively. **c** Geometric analysis of the contractile ring closure. Eccentricity and ring edge ingression are the degree of ring off-centering and the distance between the ring edge and initial ring perimeter, respectively. **d** Trajectory of ring constriction. Outlines of the contractile ring during cytokinesis are plotted based on the segmentation results. The color coding indicates time relative to cytokinesis onset. **e** Mean ring closure and eccentricity. **f** Mean ring edge ingression. Lagging edge furrowing onset was defined as the timing when the $L_{lag}(t)$ exceeded 0.02 (2% relative to initial ring size). Times are relative to cytokinesis onset. Error bands indicate 95% confidence intervals. Scale bars, 10 μm. Source data are provided in a Source Data file.

Conversely, in epithelial tissues, unilateral cytokinesis is independent of spindle positioning and is mechanically regulated. In various vertebrate and invertebrate epithelia, the mere spindle-cortex proximity does not dictate the leading edge of the ring[9,35–38]. However, the loss of components of adherens junctions leads to symmetric ring closure, at least in *Drosophila* embryonic and follicular epithelia[5,38]. These studies propose that the passive mechanical resistance imposed by adherens junctions anchors the contractile ring to the apical cell cortex, thereby allowing the ring to close from basal to apical[5]. This model explains influence of mechanical stress on the ring closure direction but does not account for the mechanical regulation of contractile ring assembly and function at the molecular level.

*C. elegans* zygotes offer a unique model system to study unilateral cytokinesis because they lack both adherens junctions and a transversely off-centered spindle[9]. In this system, contractile ring components concentrate at the leading edge, forming a structurally asymmetric ring[9]. The formation of this structurally asymmetric ring is regulated by contractile ring components anillin and septin via unknown mechanisms[9]. Recent studies using these zygotes have shown that cortical flow, a concerted movement of cell cortex components at the cell surface, facilitates contractile ring assembly through a mechanical process[39–41] (Fig. 1a). Cortical flow is driven by a gradient in cell cortical contractility[42], and during cytokinesis, it is oriented from the polar region toward the ring (hereafter called ring-directed flow). These studies proposed that ring-directed cortical flow from opposite poles compresses the gel-like cortex, thereby enhancing and aligning myosin concentration and actin orientation, respectively[39–41]. Interestingly, the amplitude of ring-directed cortical flow differs between the leading and lagging cell cortex[39], and artificial cellular compression via external forces influences both the pattern of

cortical flow and the degree of ring closure asymmetry[43]. Nevertheless, it remains unknown whether asymmetric closure requires (1) ring-directed flow, and (2) mechanical regulation, given the involvement of multiple confounding factors, compression-dependent cellular rotation occurring in a commonly-used imaging condition which prevents us from analyzing leading and lagging edges as they rotate[43], the lack of comparative 4D analysis in mutants of actomyosin regulators, and the challenges in simultaneously imaging the cell cortex and the ring.

We hypothesized that asymmetric ring closure in *C. elegans* involves asymmetric regulation of the cortical flow through mechanical processes. To elucidate the mechanism of asymmetric ring closure, we performed 4D live imaging. We developed computational tools to measure various parameters, such as ring closure, ring eccentricity, cortical flow, cortical compression, and myosin enrichment rate, from the same 4D live imaging datasets (Fig. 1b). Using these data obtained in normal and mutant cells, we gained an integrative view of mechanochemical contractile ring regulation during asymmetric ring closure. Quantitative analysis of these parameters suggests that the mechanical inhibition of the ring-directed cortical flow is associated with the furrowing delay at the lagging cell cortex. To confirm the role of ring-directed cortical flow in asymmetric contractile ring closure, we conducted an in vitro cell-bead adhesion assay in which we artificially obstructed the ring-directed cortical flow in symmetrically dividing cells. Finally, through a genetic analysis, we identified the molecular pathway that tunes contractile ring mechanosensitivity.

## Results

### High-resolution 4D imaging and analysis of contractile ring

To understand the mechanism of asymmetric ring closure, we performed high-resolution 4D imaging of endogenously tagged non-muscle myosin II::GFP (NMY-2) in *C. elegans* zygotes (Fig. 1b and Supplementary Fig. 1a). We applied a new imaging method that combines the use of spacer beads and a refractive index-matching sample medium to avoid cellular compression and to improve image resolution[44], respectively (see Methods for detail). Cellular compression is detrimental as it induces global cellular rotation[43], hindering our ability to consistently track the leading and lagging edges during division, and it is also known to induce spindle off-centering-dependent asymmetric ring closure in some species[33]. The improved imaging method allowed us to capture uncompressed embryonic volumes (30 μm thickness) at 5.6-s intervals with a 1 μm step size, without cell rotation (Supplementary Fig. 1b and Supplementary Movie 1).

We developed image analysis pipelines to quantify the dynamics of contractile ring closure, cortical flow, cortical compression, and myosin intensity using the same 4D live imaging dataset (Fig. 1b). First, we analyzed the dynamics of contractile ring closure by performing image segmentation of the contractile ring (Fig. 1b and Supplementary Fig. 1c). Using the segmentation data, we measured the ring radius and ring center coordinates at every time point. From these measurements, we calculated the ring closure indices $R_{closure}(t)$ (with 1.0 representing complete closure), ring eccentricities $E(t)$ (indicating off-center tendency), and the degrees of leading-edge and lagging-edge ingression $L_{lead}(t)$, $L_{lag}(t)$ (Fig. 1c, see Methods). Due to natural variations in cytokinesis timing, we aligned the time series data relative to the time around 10% contractile ring closure (Supplementary Fig. 1d). We also defined cytokinesis onset based on the extrapolation of an initial linear part of the ring closure curve to 0 (Supplementary Figs. 1d and 2). Markedly, the newly developed imaging and quantification methods allowed us to detect differences in sub-percentage ring closure indices (Fig. 1d–f). The data obtained show that the contractile ring initially undergoes nearly the upper limit of physically possible asymmetric closure until the start of lagging edge furrowing (Fig. 1d–f; Supplementary Movie 1, when $R_{closure}(t) \approx E(t)$). These data imply that the primary factor contributing to ring off-centering is the time interval between the onsets of furrowing at the leading edge and the lagging edge.

### Identification of causes underlying asymmetric ring closure

To quantitatively understand factors contributing to asymmetric ring closure, we analyzed the relationship among obtained parameters, such as peak eccentricity, time lag between leading and lagging edge ingression, and ring closure velocity. The time lag, Δt, was determined by the delay of lagging edges reaching 10% ingression relative to the initial ring diameter (Fig. 1f). Additionally, ring closure velocity, $v$, was calculated based on the slope of the $R_{closure}$ curve (a linear part spanning $0.2 < R_{closure} < 0.7$ in Fig. 1e). We found that ring closure velocity and peak eccentricity have no clear correlation (Pearson's $r = 0.352$, $p = 0.028$, $n = 39$; Fig. 2a, left). On the contrary, the time lag is weakly correlated with the peak eccentricity (Pearson's $r = 0.54$, $p = 0.0003$, $n = 39$; Fig. 2a, middle). We also considered the degree of time lag relative to the ring closure speed, defined as the normalized time lag $\Delta t_n$, by multiplying the ring closure velocity by the time lag (Fig. 2a, right). We found that the normalized time lag is strongly correlated with peak eccentricity (Pearson's $r = 0.93$, $p < 0.0001$, $n = 39$), indicating that both ring closure velocity and time lag control asymmetric ring closure in normal cells.

Next, we analyzed the potential causes of unilateral cytokinesis defects in mutants or RNAi knockdown of actomyosin regulators (a list of genes is in Fig. 2b and Supplementary Fig. 3a). Note that our knockdown condition is mild for some genes, allowing cytokinesis completion. First, we analyzed upstream of RhoA, such as RhoA activators and inhibitors. We found that ECT-2/RhoGEF is required for asymmetric ring closure (Fig. 2c, d). Conversely, RhoGAP and a worm specific RhoA activator NOP-1 are dispensable for unilateral cytokinesis (Fig. 2d and Supplementary Fig. 3b). Second, we analyzed RhoA effectors. Knockdown of ROCK increased peak eccentricity (Fig. 2c, d). On the other hand, knockdown of anillin and septin resulted in symmetric ring closure (Fig. 2c, d and Supplementary Fig. 3b). Additionally, we found that knockdown of myosin regulatory light chain kinase/MRLC and gain-of-function mutation of actin gene *act-2*[45] reduced ring eccentricity, whereas formin knockdown did not (Fig. 2d and Supplementary Fig. 3b). Finally, we analyzed a pathway of branched actin network formation, which is known to inhibit myosin-dependent actin-contractility[46,47]. Arp2 is a component of the Arp2/3 complex, which regulates branched nucleation of actin from the existing filaments[48]. On the other hand, Coronin removes Arp2/3 from actin, promoting actin debranching[49–51]. Although Arp2 knockdown exhibited a normal level of eccentricity, we found that Coronin knockdown resulted in symmetric ring closure. Thus, our analysis confirmed previously reported unilateral cytokinesis phenotypes for anillin[9], septin[9], ROCK[9], NOP-1[52], and Arp2[53], with providing more quantitative information, and identified novel roles of ECT-2, Coronin, actin, and MRLC in unilateral cytokinesis.

Our data show that knockdown of actomyosin regulators with unrelated molecular functions causes the same phenotype. This phenomenon is known as "degeneracy", and a previous study also showed that degeneracy makes it difficult to intuitively estimate the roles of actomyosin regulators in cortical dynamics from their biochemical function[54]. Therefore, we prioritized data-driven hypothesis formulation by comparing the obtained parameters. Time lag (raw values) and peak eccentricity were not particularly correlated (Pearson's $r = 0.57$, $p = 0.04$; Supplementary Fig. 3c). However, we observed a strong correlation between normalized time lag and peak eccentricity (Pearson's $r = 0.99$, $p < 0.0001$; Fig. 2e and Supplementary Fig. 3d). Furthermore, when examining the relationship between ring closure velocity and time lag, we found that conditions with asymmetric and symmetric closures are separated within the parameter space (Fig. 2f). Cells exhibiting slower ring closure, as well as shorter time lag, tended to display defective unilateral cytokinesis.

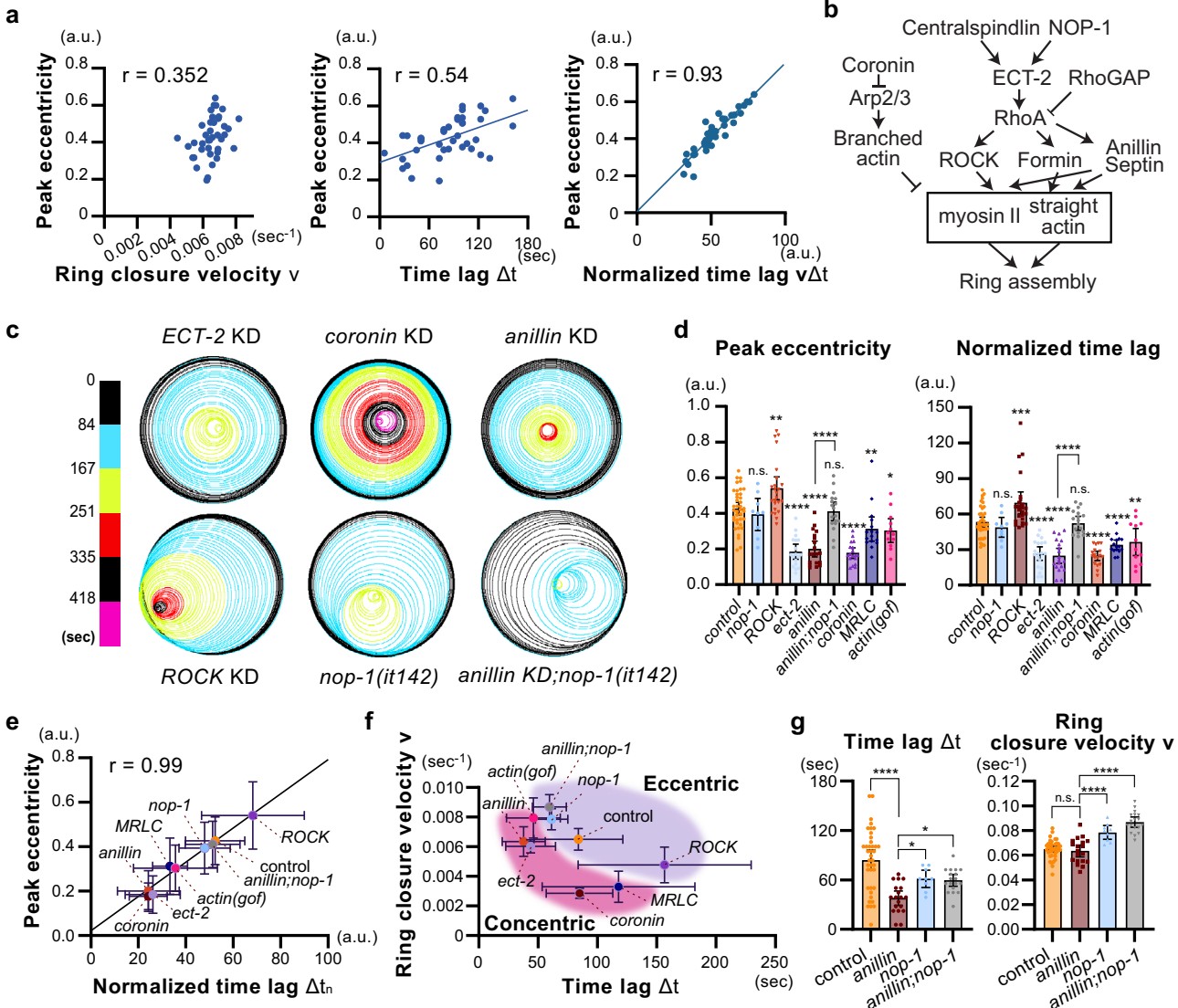

**Fig. 2 | Ring closure velocity and a delay in lagging edge furrowing collectively determine the ring eccentricity. a** Relationship among peak eccentricity, ring closure velocity, time lag, and normalized time lag in normal cells. Normalized time lag was calculated by multiplying ring closure velocity and time lag. **b** Cell cortex components analyzed in this study (see text). **c** Ring trajectories of different RNAi and mutant backgrounds. **d** Mean peak eccentricity and normalized time lag in different backgrounds. **e** Relationship between mean peak eccentricity and mean normalized time lag in different backgrounds. **f** Relationship between mean ring closure velocity and mean time lag in different backgrounds. **g** Mean time lag and mean ring closure velocity in anillin KD. Times are relative to cytokinesis onset. Error bars indicate 95% confidence intervals except for panel E and F, in which standard deviations were shown. The value *r* represents Pearson's correlation coefficient. *P*-values were calculated using one-way ANOVA followed by Holm–Sidak's multiple comparison test. Scale bars, 10 µm. Source data are provided in a Source Data file.

Anillin is a scaffolding protein that interacts with RhoA, actin, myosin, and septin during cytokinesis, facilitating contractile ring assembly[55,56]. Although anillin knockdown leads to symmetric ring closure, the phenotype can be fully rescued by a *nop-1* mutation[52]. This previous observation makes the role of anillin in unilateral cytokinesis mysterious. We found that the *nop-1* mutation increased both time lag and ring closure velocity compared to *anillin* KD alone (Fig. 2g), suggesting that the rescue is due to the increase in these parameters. Consistently, we newly found that the septin KD phenotype was also rescued by the *nop-1* mutation, accompanied by an increase in ring closure velocity (Supplementary Fig. 3e). These results, obtained using control, RNAi, and mutant embryos, all suggest that ring eccentricity is collectively determined by the ring closure velocity and a delay in furrowing at the lagging edge.

## Cell cortex exhibits orthogonally directed cortical flow

We next analyzed cortical flow as it might relate to a relative furrowing delay at the lagging edge. As reported previously[39], we observed ring-directed cortical flow at both the leading and lagging edges, with a delay at the lagging side (Fig. 3a and Supplementary Movie 2). To quantitatively measure cortical flow dynamics, we employed particle image velocimetry (PIV)[57] (Fig. 4a and Supplementary Fig. 4, and Supplementary Movie 3), and confirmed that ring-directed flow is indeed delayed at the lagging cell cortex (Fig. 4c; dotted lines). During this delay, we identified a previously uncharacterized, orthogonally oriented flow, which we termed circumferential cortical flow (Fig. 3a, bottom panel and Supplementary Movie 2). Notably, circumferential flow does not occur at the leading edge and is distinct from the previously reported rotational flow observed when the cell is subjected to external compressive forces (Fig. 3a and Supplementary Fig. 1b). In the

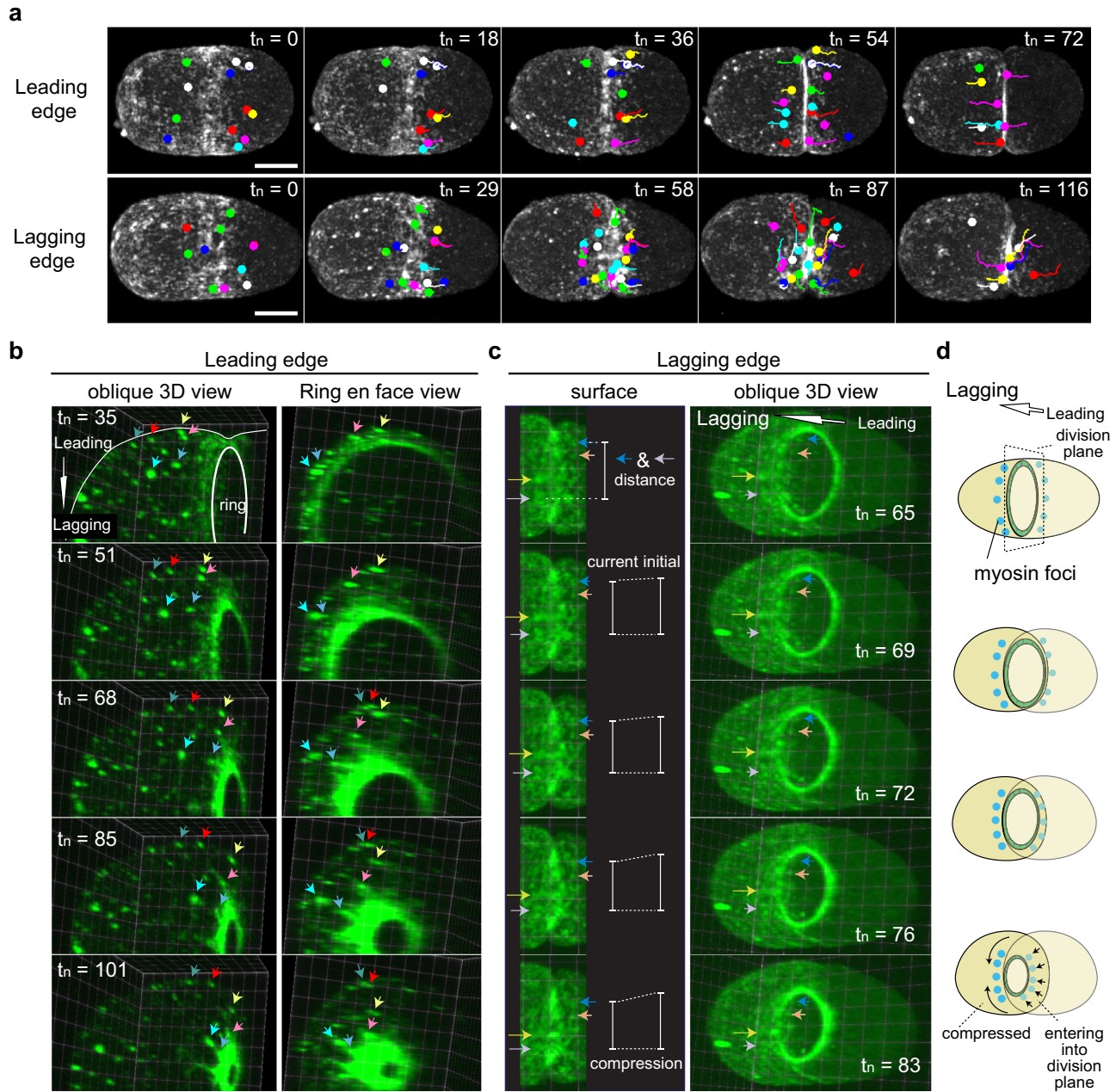

**Fig. 3 | Cell cortex exhibits orthogonally directed cortical flow during asymmetric ring closure. a** Tracking of myosin foci during cytokinesis. $t_n$ represents normalized time, derived by multiplying time and ring closure velocity. **b** 4D tracking of ring-directed cortical flow. MRLC knockdown embryos were used due to their persistent and large myosin foci (see Supplementary Movie 1 for control RNAi). Each differently colored arrow tracks a unique myosin focus over time. The oblique 3D view and ring en face view were obtained from the identical embryos, with the former showing both the cell surface and the division plane. **c** 4D tracking

of circumferential cortical flow. Control RNAi embryos were used. In the left panels, distance between myosin foci were measured over time, and their incremental changes were shown by white vertical lines. **d** Summary of cortical flow during asymmetric ring closure. At the leading edge, ring-directed flow delivers cortical materials from the polar region into the division plane. These myosin foci eventually join the contractile ring. At the lagging edge, circumferential compressional flow is generated as the ring constricts. Times are relative to cytokinesis onset. Scale bars, 10 μm. Source data are provided in a Source Data file.

following analyses, we characterized the mechanisms and functions of these orthogonally oriented flows.

## Mechanism and function of ring-directed flow

The roles and mechanisms of ring-directed cortical flow have not been fully characterized. It has been proposed that ring-directed flow contributes to the positive feedback regulation of contractile ring assembly (Fig. 4b)[39]. According to this model, ring-directed cortical flow delivers cortical material to the equator, enriching myosin through ring-directed compression. The enriched myosin, in turn,

promotes ring constriction, which is believed to further facilitate ring-directed flow. This model finds support in the correlational exponential increase of three parameters over time: ring-directed compression, myosin intensity in the ring, and ring closure velocity. However, the causal relationships among these factors remain unclear. Specifically, two key questions persist: (1) whether the cortical materials delivered to the equator are indeed utilized by the contractile ring, and (2) what drives ring-directed flow. The fate of delivered cortical materials was previously speculated based on the turnover of ring myosin[39], but direct observation has been hindered by technical difficulties in

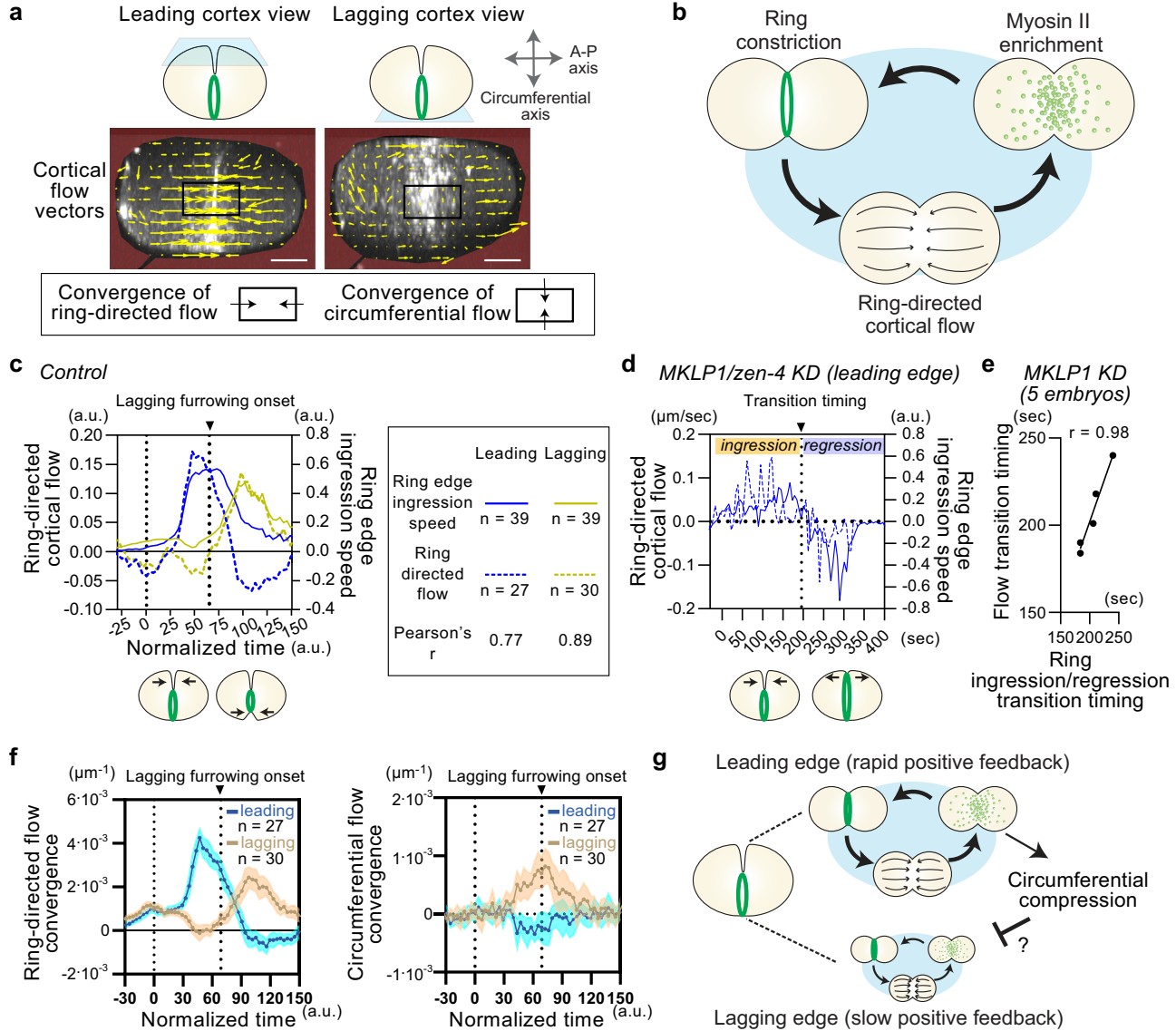

**Fig. 4 | Ring-directed flow is generated by ring closure but is inhibited at the lagging cell cortex. a** Cortical flow measurement. Leading and lagging cortex views were derived from 4D imaging data. Cortical flow vectors were estimated using particle image velocimetry (yellow arrows). Mean cortical flow velocities in the area immediately anterior to the ring were used to calculate ring-directed flow. Convergence of cortical flow vectors was also derived to estimate the influx of flow within the equatorial ROI (black boxes). **b** Cortical flow-dependent positive feedback model in contractile ring assembly. Ring-directed cortical flow delivers cortical materials to the equator, enriching myosin and promoting ring constriction. Ring constriction likely triggers ring-directed flow, as suggested by previous in silico analyses. Relationship between ring edge ingression speed and ring-directed cortical flow. The leading and lagging edge of control RNAi (**c**, mean value) and the leading edge of *zen-4*/MKLP1 RNAi were shown (**d**, single embryo). **e** Correlation between transition timings of ingression/regression and ring-directed/ opposing flow. **f** Mean convergence of ring-directed and circumferential flow. **g** Potential mechanism of ring-directed flow inhibition at the lagging edge (See texts). Times are relative to cytokinesis onset. Error bands indicate 95% confidence intervals. Scale bars, 10 μm. Source data are provided in a Source Data file.

visualizing myosin foci in the division plane. Our high-resolution 4D dataset captures the movements of myosin foci in the division plane of normal cells (Supplementary Movie 1). In MRLC knockdown, which exhibits more persistent and larger myosin foci than the control, we were able to track myosin foci as they migrated from the polar region of the cell surface into the division plane, eventually joining the contractile ring (Fig. 3b and Supplementary Movie 4). Thus, our results suggest that ring-directed flow delivers cortical materials to the contractile ring.

Next, we examined the mechanisms responsible for generating ring-directed flow. Cortical flow is driven by a gradient in actomyosin contractility[42], and a computer simulation have indicated that higher contractility at the equator, compared to the pole, drives ring-directed flow[58]. Indeed, inhibition of cortical contractility at the polar cell cortex

facilitates ring-directed flow[25,27]. Nevertheless, the relationship between ring-directed flow and ring closure has remained unclear due to the absence of 4D analysis. We plotted velocities of ring edge ingression (Fig. 4c; solid lines) and ring-directed cortical flow (Fig. 4c; dotted lines) and found that they correlate well both at the leading and lagging edges (leading edge: Pearson's $r = 0.77$, $p < 0.0001$, lagging edge: Pearson's $r = 0.89$, $p < 0.0001$). We also compared the velocity of ring edge ingression and ring-directed flow in *zen-4*/MKLP knockdown. MKLP is a component of centralspindlin, and its knockdown leads to ring regression midway through cytokinesis. We observed ring-directed flow during ring constriction; however, when the ring started to regress, cortical flow moved away from the ring (Fig. 4d). The timing of transition between ingression and regression of the ring strongly correlated with transitions between ring-directed and pole-

ward flow (Pearson's $r = 0.98$, $p < 0.0001$; Fig. 4e). Therefore, we conclude that the invaginating ring edge consistently pulls the nearby cell cortex, generating ring-directed cortical flow. These findings shed light on the origin and function of ring-directed flow and provide further support for the flow-dependent positive-feedback model in ring assembly.

## Mechanism and function of circumferential cortical flow

At the lagging edge, myosin foci converge as they move circumferentially towards the center (Fig. 3c; left panel, and Supplementary Movie 5). A 3D visualization demonstrates that circumferential flow is generated when the ring constricts but does not ingress from the lagging edge (Fig. 3c; right panel, and Supplementary Movie 5). These results suggest that the ring constriction influences the lagging edge differently from the leading edge; it does not pull the polar cortical materials but rather compresses the lagging cell cortex.

Quantitative analysis also supports these observations. To better measure the flow dynamics at the equatorial region, we calculated the convergence of flow vectors along different axes (Fig. 4a). Here, when the overall influx of flow in the equatorial region is greater than the efflux, we will obtain a positive convergence value. At the leading cell cortex, the convergence of ring-directed flow increased rapidly and reached its peak before the onset of lagging edge furrowing (Fig. 4f; left). Conversely, the convergence of ring-directed flow in the lagging cell cortex decreased shortly after cytokinesis onset and reached a local minimum before the onset of lagging edge furrowing (Fig. 4f; left, Fig. 3e; $t_n = 58$). These results indicate that ring-directed flow is inhibited at the lagging cell cortex. During this period, we observed an increase in circumferential axis convergence at the lagging cell cortex (Fig. 4f; right). Taken together, these results indicate that ring closure induces circumferential cortical flow at the lagging edge, which coincides with the inhibition of ring-directed flow.

## Cortical compression suppresses ring-directed cortical flow

As ring-directed flow is driven by ring closure, one plausible mechanism that inhibits ring-directed flow at the lagging cell cortex is the tug-of-war between the contractile cell cortex and the constricting ring. Therefore, we hypothesized that circumferential compression inhibits ring-directed flow, thereby suppressing a flow-dependent positive feedback loop in ring assembly (Fig. 4g). To test the relationship between cortical compression and ring-directed flow, we induced ectopic cortical compression using a semi-dominant temperature-sensitive mutation of actin, *act-2(or295)*, known to cause actin filament stabilization and ectopic cortical contractility[45]. In normal cells, ring-directed flow at the lagging edge is interrupted by a "no-flow" period (Fig. 5a, b). In *act-2(or295)* cells, the number of no-flow periods increases (Fig. 5a). By tracking myosin foci during cortical contraction and relaxation, we found that ring-directed flow was suppressed during contraction and induced during relaxation (Fig. 5c, d). Ring closure is necessary for the relaxation-induced ring-directed cortical flow, as similar flow was not observed post-cytokinesis (Fig. 5d; right). These results suggest that non-ring-directed cortical compression inhibits ring-directed cortical flow at the lagging cell cortex (Fig. 5e).

## Cortical compression inhibits ring edge ingression

We next analyzed the effects of cortical compression on ring closure dynamics. A previous study demonstrated that laser-induced cuts of the cell cortex adjacent to the ring did not accelerate the ring closure rate[39]. This suggests that ring closure is not constrained by the mechanical resistance of the cell cortex connected to the ring. If this is indeed the case, cortical compression would not limit ring closure, and therefore, is not required for ring closure asymmetry. We tested this possibility using highly contractile *act-2(or295)* mutants and found that the overall ring closure curves were similar to those of the control (Fig. 6a), with the ring velocity even exceeding that of the control

(Fig. 2f). Thus, as predicted, the overall ring constriction rate is not limited by cortical compression. However, the contractile rings of the *act-2* mutant frequently displayed oscillatory movement during constriction and exhibited a lower mean peak eccentricity, likely due to wandering (Figs. 6a and 2d). These results suggest that cortical compression influences ring position without altering the constriction rate.

We investigated how cortical compression affects ring positioning. To infer cortical compression, we measured myosin intensity changes at the cell surface closest to the objective lens (referred to as the top surface) (Fig. 6b). The upward and downward movements of contractile ring along the imaging axis were also measured using the same samples (Fig. 6b). During contraction at the top surface, the downward movement of the top ring edge was often halted (Fig. 6c–e; labeled as "contraction," Supplementary Movie 6). Conversely, relaxation of the top surface often coincided with the downward movement of the top ring edge (Fig. 6c–e; labeled as "relaxation," Supplementary Movie 6). Quantitative analysis of cortical compression and acceleration of the top ring edge also confirmed these trends (Fig. 6f, g). These findings suggest that cortical compression adjacent to the ring impedes the ingression of the nearby ring edge.

How does cortical compression inhibit ring edge ingression without delaying the overall ring constriction rate? Notably, we frequently observed the upward movement of the bottom ring edge when the top surface is compressed (Fig. 6c–e, h). Thus, it is likely that when a certain part of the ring edge ingression is prevented, other areas of the ring edge pull more cortex from the nearby surface, maintaining a similar constriction rate.

## Manipulation of ring-directed flow causes asymmetric closure

Our data suggest that cortical compression inhibits ring-directed flow and ring edge ingression, but it is unclear if they constitute a single pathway or redundant mechanisms (Fig. 6i). To directly test the role of ring-directed flow in ring positioning, we inhibited ring-directed flow in symmetrically dividing cells. It is known that the two-cell stage AB cell undergoes asymmetric ring closure, similar to the $P_0$ zygote[59] (Fig. 7a and Supplementary Fig. 5a). We found that unilateral ring closure requires cell contact with the neighboring $P_1$ cell (Fig. 7b, c). To inhibit ring-directed cortical flow, we attached adhesive polystyrene beads to the symmetrically dividing isolated AB cell (Fig. 7d, e). The beads, coated with positively charged Rhodamine fluorescent dyes and salts, adhere to the plasma membrane. This adhesion is presumed to occur through electrostatic interaction, similar to the mechanism observed with poly-L-lysine coating. As shown in our previous study, attachment of the 30 μm diameter beads reduced ring-directed flow in the area proximal to the bead attachment site (Fig. 7e)[4]. The response scaled with the bead diameter, indicating that passive mechanical resistance imposed by adhesive beads limits ring-directed flow (Fig. 7e). A single bead attachment resulted in a slight increase in the average peak eccentricity, although this change was not statistically significant (Fig. 7c, $p = 0.11$). This result may be attributed to the smaller contact area compared to in vivo conditions (Fig. 7a, d) and the normal ring-directed flow at the cell cortex distal to the bead (Fig. 7e). Notably, the attachment of the two 30 μm beads increased peak eccentricity, and about 50% of the cells exhibited a level of ring eccentricity comparable to that of the intact AB cell in vivo (Fig. 7c, f–h). Thus, we conclude that mechanical obstruction of ring-directed cortical flow is sufficient to induce asymmetric ring closure.

## Anillin tunes mechanosensitivity of ring-directed flow

Our analysis revealed that asymmetric ring closure results from contractile ring mechanosensation, which involves the mechanical inhibition of ring closure-dependent generation of ring-directed flow (See model in Fig. 9g; cortical flow pathway and mechanical cue). Consistently, the lack of circumferential compression, as measured by the deformation rate of the lagging cortex, resulted in symmetric ring

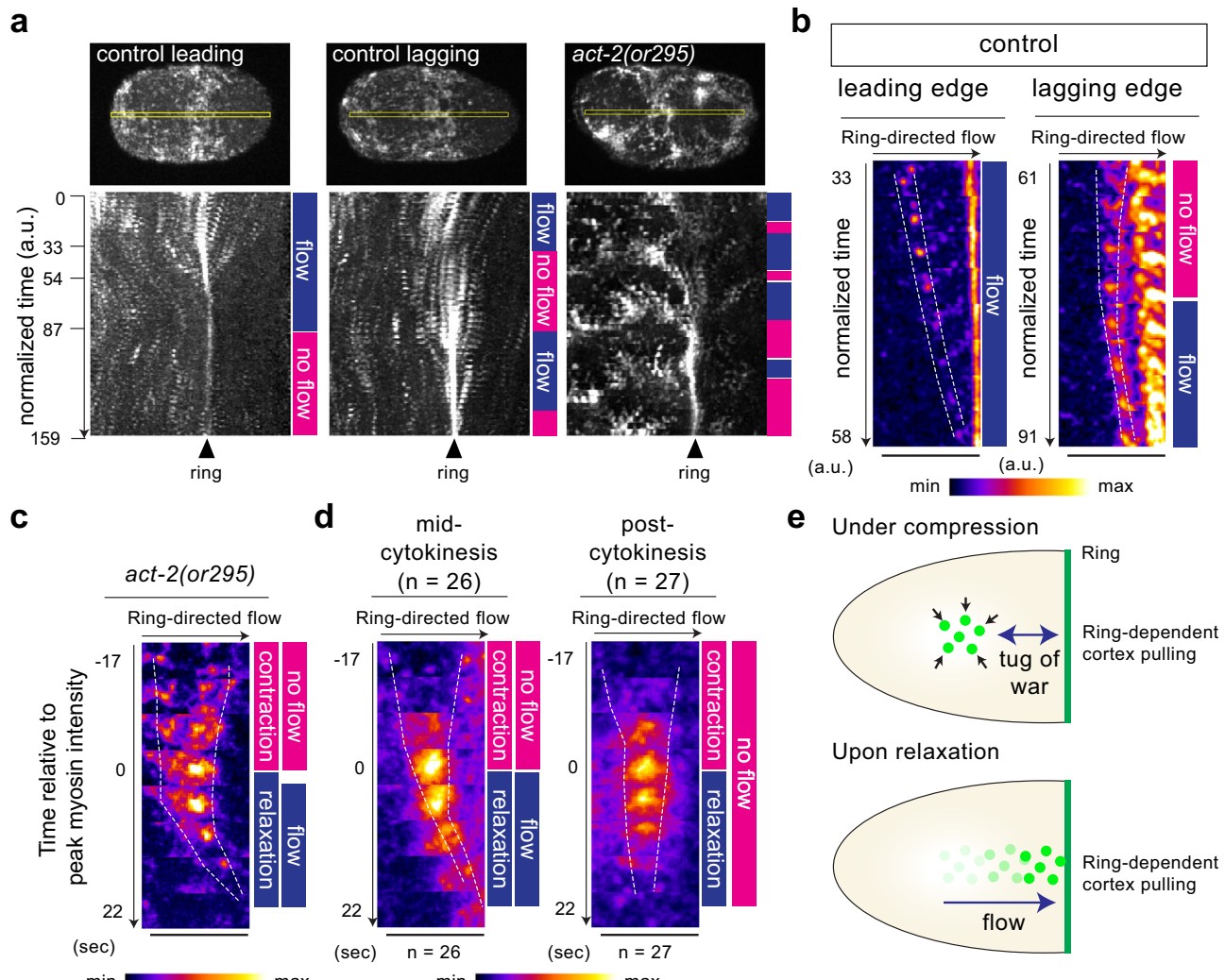

**Fig. 5 | Cortical compression suppresses ring-directed cortical flow.**
**a** Kymographs of cortical myosin. Yellow boxes in the top images were used to generate kymographs. Periods of ring-directed flow and no flow were indicated by colored boxes next to the kymographs. *act-2(gof)* exhibits high cortical contractility and increased no-flow periods. **b** Ring-directed cortical flow in control zygotes. The leading edge exhibits consistent flow, while lagging edge flow is interrupted by a no-flow period. **c** Ring-directed cortical flow in *act-2(gof)* mutants. Movement of a myosin focus during contraction and relaxation of the region immediately anterior to the ring is shown. **d** Averaged images of myosin foci during cortical contraction and relaxation in *act-2(gof)* mutants. **e** Mechanism of compression-induced inhibition of ring-directed cortical flow. Ring-directed flow is impeded by a tug-of-war between the constricting ring and the contractile cortex. Times are relative to cytokinesis onset. Scale bars, 10 μm. Source data are provided in a Source Data file.

closure in zygotes of *ect-2*, MRLC, and Coronin knockdown (Figs. 8c and 2d). However, there is one exception: knockdown of anillin resulted in symmetric closure even though embryos retained a normal level of circumferential compression (Figs. 8c and 2d). Visualization of cortical flow at the lagging edge demonstrated that ring-directed flow is not interrupted in anilin KD, but the phenotype was rescued by the *nop-1* mutation (Fig. 8a).

To characterize spatio-temporal changes in cortical mechanics, we measured cortical convergence of flow vectors. Convergence and divergence of flow at the surface inform cortical compression and relaxation, respectively, provided there is no involution or extrusion[60]. In control embryos, we observed that the convergence of ring-directed flow at the lagging cell cortex decreased upon cytokinesis onset, indicating cortical relaxation (Fig. 8b; top left). However, the relaxation was not apparent in anillin KD (Fig. 8b; top middle). The relaxation reemerged with the introduction of a *nop-1* mutation (Fig. 8b; top right). The rate of relaxation, estimated from the lines connecting local maxima and minima of flow convergence (red and blue arrowheads in the Fig. 8b), also confirmed these observations (Fig. 8d). On the other

hand, there were no marked differences in circumferential axis compression under these conditions (Fig. 8b; bottom row, and Fig. 8c). These data suggest that anillin is required for the mechanosensitive inhibition of ring-directed cortical flow (Fig. 8e). Consistently, anillin KD also reduced ring eccentricity during the presumable adhesion-dependent asymmetric ring closure of the two-cell stage AB cell, whereas Coronin KD underwent normal asymmetric ring closure (Supplementary Fig. 5b). Thus, anillin is specifically required for contractile ring mechanosensation.

**Anillin maintains flow-dependent feedback in ring assembly**
We next aimed to uncover the mechanism underlying defective ring mechanosensation in anillin KD. Since cortical flow plays essential roles in normal ring mechanosensation, we analyzed the function of a cortical flow-dependent positive-feedback loop in ring assembly (Fig. 4b). We observed a strong correlation between ring closure velocity and peak ring-directed cortical flow across different RNAi/mutant groups, including anillin KD (Fig. 9a: Pearson's $r = 0.94$, $p = 0.0002$). This further confirms the critical roles of ring constriction

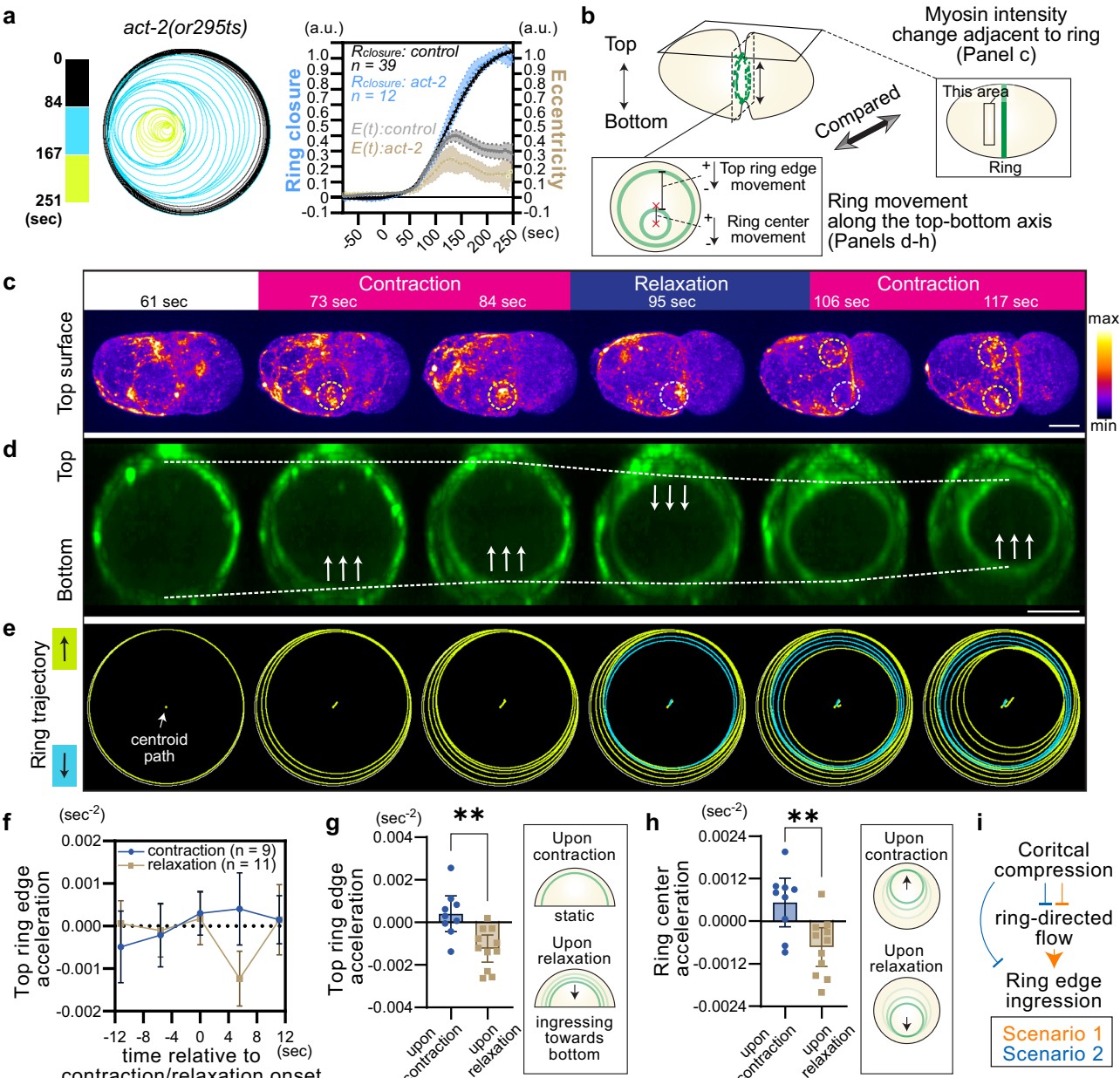

**Fig. 6 | Cortical compression delays ring edge ingression without suppressing ring constriction. a** Contractile ring trajectory of a single *act-2(or295ts)* mutant (left panel) and the mean values of ring closure and eccentricity (graph). **b** Explanation of experiments related to panels C-H. Changes in myosin intensity adjacent to the ring were measured to monitor cortical contraction and relaxation. Ring movement along the top-bottom axis during contraction/relaxation was also analyzed. **c** MyosinII::GFP at the top surface during cytokinesis. Yellow and light blue dotted circles indicate areas exhibiting contraction and relaxation, respectively. **d** Ring en face view of the same embryo shown in panel C. Dotted lines indicate the top and bottom edges of the contractile ring. Arrows indicate the ingression of the ring edge. **e** Ring trajectories of the contractile ring shown in panel D. Yellow and blue rings indicate upward and downward movements, respectively. **f** Mean acceleration of the top ring edge during cortical contraction and relaxation. **g** Mean acceleration of the top ring edge measured 5.6 s after contraction and relaxation. **h** Mean acceleration of the ring center measured 5.6 s after contraction and relaxation. **i** Possible regulatory mechanisms indicated by data in Figs. 5 and 6. Error bars and bands indicate 95% confidence intervals. Scale bars, 10 μm. Source data are provided in a Source Data file.

in the generation of ring-directed flow. Additionally, we found that the myosin enrichment rate is highly correlated with ring closure velocity and peak ring-directed cortical flow among sample groups, excluding anillin KD (Fig. 9b: Pearson's $r = 0.88$, $p = 0.004$, Fig. 9c: Pearson's $r = 0.84$, $p = 0.009$). In contrast, anillin KD exhibited an excess myosin enrichment rate relative to the ring closure velocity and peak ring-directed cortical flow, as evidenced by studentized residuals greater than the conservative threshold for outliers of 3 (Fig. 9b, c). These results suggest that the positive feedback loop between ring-directed

cortical flow, myosin enrichment rate, and ring closure was disrupted in anillin KD embryos.

The myosin enrichment rate used for the above analysis was calculated based on the temporal change in the total level of ring myosin (Supplementary Fig. 6). A previous study reported that anillin is required for the asymmetric enrichment of myosin during unilateral cytokinesis[9]. However, a simple loss of asymmetry would still preserve the total myosin signal intensity. Contrary to this prediction, we observed an exceptionally high level of total ring myosin midway

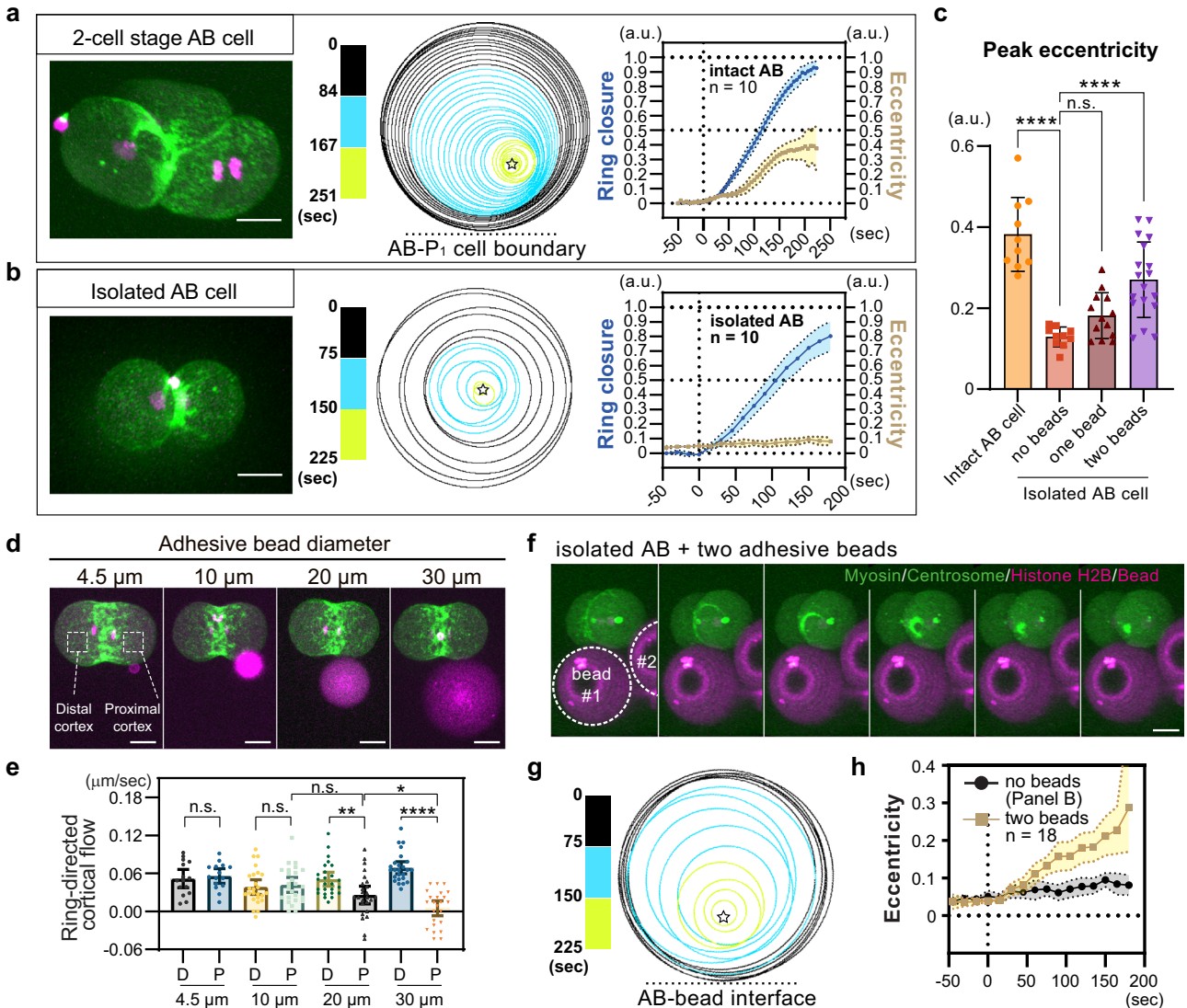

**Fig. 7 | Artificial inhibition of ring-directed cortical flow is sufficient to induce asymmetric ring closure.** Asymmetric ring closure in the AB cell of intact two-cell stage embryos (**a**) and symmetric ring closure in the isolated AB cell (**b**). Myosin II::GFP and mCherry::histone H2B (left), ring trajectories (middle), and mean ring closure and eccentricity curves (right) are shown. **c** Mean peak eccentricity of the contractile ring in intact and manipulated AB cells. **d, e** Artificial obstruction of ring-directed flow using adhesive beads. Attachment of adhesive beads of different sizes to the isolated AB cell. Mean ring-directed flow in areas distal and proximal to the bead, as shown in (**d**), was measured in (**e**). **f–h** Induction of asymmetric ring closure due to bead attachment. 3D Oblique view of contractile rings in the bead-attached AB cell (**f**). Ring trajectory and mean eccentricity are shown in g and h, respectively. Times are relative to cytokinesis onset. Error bands and bars indicate 95% confidence intervals. Scale bars, 10 μm. Source data are provided in a Source Data file.

through cytokinesis (Fig. 9d–f). The abnormal enrichment of myosin was rescued by the *nop-1* mutation, indicating that the restoration of contractile ring mechanosensitivity is due to the rescue of the myosin enrichment rate (Fig. 9d–f). Our analysis also demonstrates that the total level of myosin does not dictate ring closure speed (Fig. 9b, e; e.g., control vs Coronin vs ROCK). Instead, the myosin enrichment rate outperforms in dictating ring closure velocity in samples excluding anillin KD (Fig. 9b, f). Taken together, these results suggest that a cortical flow-dependent positive feedback loop in ring assembly and maintenance of myosin enrichment rate relative to ring closure velocity and ring-directed cortical flow are critical for the mechanosensation of the contractile ring.

## Discussion

In this study, we have identified the mechanisms underpinning contractile ring mechanosensation and the tuning of mechanosensitivity. A previous study proposed a cortical flow-dependent positive feedback loop in ring assembly[39]. Our study revealed cortical flow in the division plane and further characterized the mechanism and function of ring-directed flow to strengthen this model. Notably, we newly identified circumferential compressional flow generated only at the lagging edge of the cell cortex. Through 4D quantitative analysis of a gain-of-function mutant of actin, we have demonstrated that cortical compression inhibits ring-directed flow and ring-ingression. Furthermore, our study shows that the artificial obstruction of ring-directed cortical flow by adhesive beads is sufficient to induce asymmetric ring closure in symmetrically dividing cells. Thus, our study unambiguously demonstrates that mechanical inhibition of ring-directed flow induces unilateral cytokinesis. Finally, we have also identified a molecular pathway involving anillin and *nop-1* in the regulation of ring mechanosensitivity. Ring mechanosensitivity relies on a normal rate of myosin enrichment rate within the ring, in relation to the velocities of ring closure and ring-directed cortical flow.

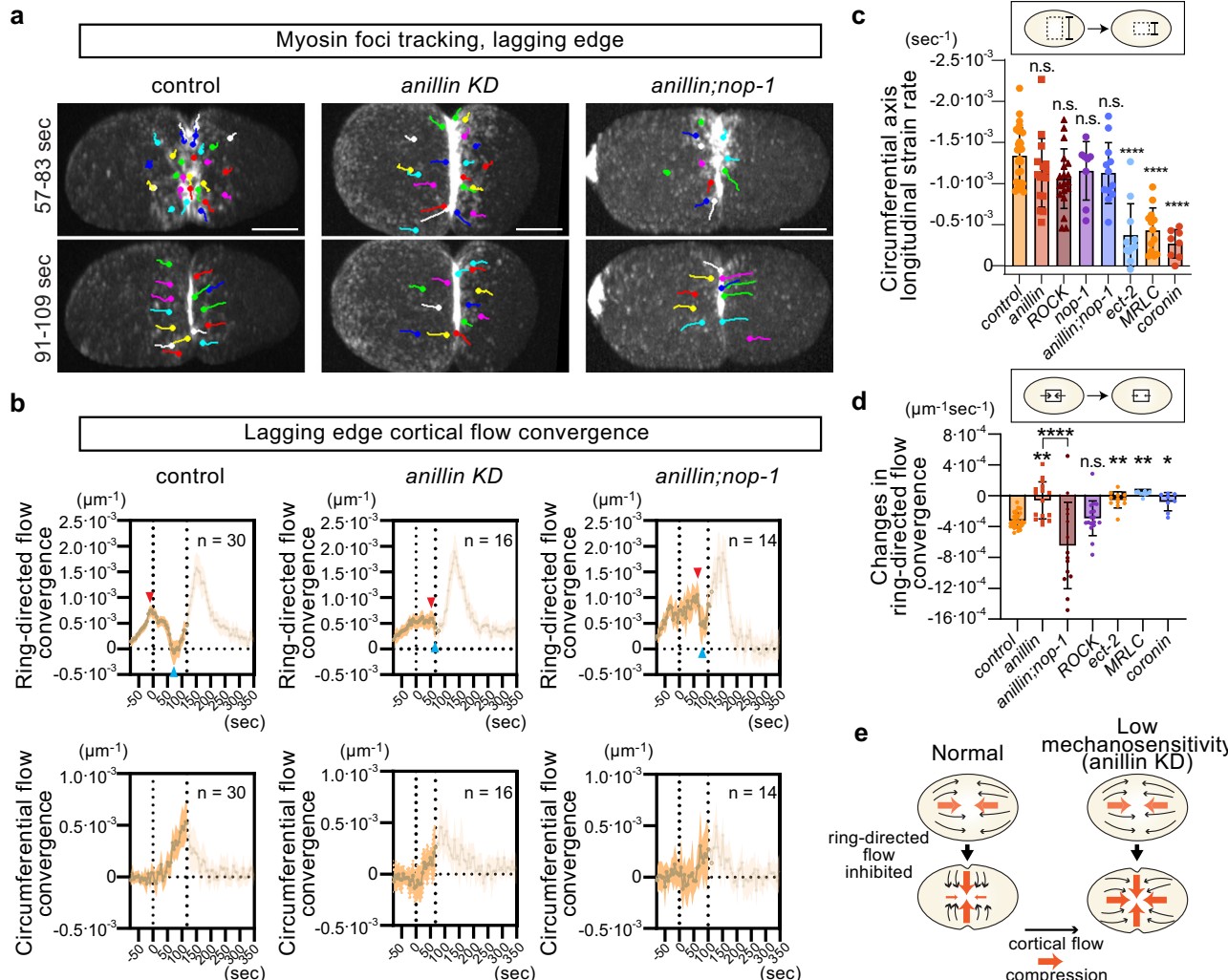

**Fig. 8 | Anillin is required for mechanosensitive inhibition of ring-directed cortical flow. a** Tracking of myosin foci at the lagging cell cortex. Symmetrically dividing anillin KD exhibits a lack of impedance of ring-directed flow, unlike control and *anillin;nop-1* (57–83 s). **b** Cortical flow convergence at the lagging edge. Mean cortical flow convergence before the onset of lagging edge furrowing is shown to monitor cortical compression. The transparent parts of graphs show the convergence after the lagging edge furrowing and are not useful for estimating cortical compression. Red and blue arrowheads indicate local maxima and minima between cytokinesis onset (left dotted line) and lagging edge furrowing onset (right dotted line) in each graph. **c** Mean longitudinal strain rate along the circumferential axis. This directly measures the compression rate. **d** Changes in convergence of ring-directed flow. Symmetrically dividing rings exhibit a lack of impedance of ring-directed compression and flow. **e** Mechanosensitive response of ring-directed flow in normal cells and its failure in anillin KD. *P*-values were calculated using one-way ANOVA followed by Holm-Sidak's multiple comparison test. Times are relative to cytokinesis onset. Error bands and bars indicate 95% confidence intervals. Scale bars, 10 μm. Source data are provided in a Source Data file.

Based on our analysis and insights from previous studies, we propose a cortical flow-dependent contractile ring mechanosensation model (Fig. 9g). In this model, locally generated mechanical cues, such as circumferential compression in zygotes and adhesive beads in our in vitro experiments, act to limit ring constriction-dependent cell cortex-pulling at the lagging cell cortex (Fig. 9g; arrows from the yellow boxes). Consequently, ring-directed cortical flow is locally suppressed at the lagging edge. This local suppression of ring-directed cortical flow should reduce the rate of flow-dependent myosin delivery to the equator, which aligns with the formation of the structurally asymmetric ring observed in previous studies[9]. Subsequently, the local reduction in myosin enrichment rate should slow down ring edge ingression at the lagging edge, leading to asymmetric ring closure. In zygotes, a subtle time lag still exists between the leading and lagging edge furrowings after the knockdown of any of the actomyosin regulators analyzed, including anillin (Fig. 2f). Therefore, we favor a scenario where this subtle asymmetry arises from the spontaneous inhomogeneity of cortical activities (Fig. 9h). Once the leading and

lagging edges are specified, ring closure compresses the lagging cell cortex, inducing circumferential flow. This, in turn, leading to the mechanosensitive inhibition of ring-directed flow. Contractile ring mechanosensation amplifies the initial subtle asymmetry, ultimately resulting in unilateral cytokinesis (Fig. 9h).

Our analysis suggests that components of the RhoA pathway tune contractile ring mechanosensitivity (Fig. 9g; left box). We demonstrate impaired mechanosensitivity in anillin KD embryos, in which myosin was too rapidly enriched in the contractile ring (Fig. 9). This observation is somewhat counterintuitive, considering the role of anillin as a scaffolding protein for actomyosin components and its involvement in the formation of large cortical patches, containing myosin[61,62]. Recent research has shown that Anillin-binding proteins GCK-1 and CCM-3 negatively regulate RhoA activity by promoting RhoGAP cortical localization[63]. Therefore, one possibility is that anillin KD relieved this negative regulation, leading to the activation of RhoA-dependent myosin enrichment. Alternatively, anillin may directly inhibit the myosin enrichment rate, independently of its positive role in cortical

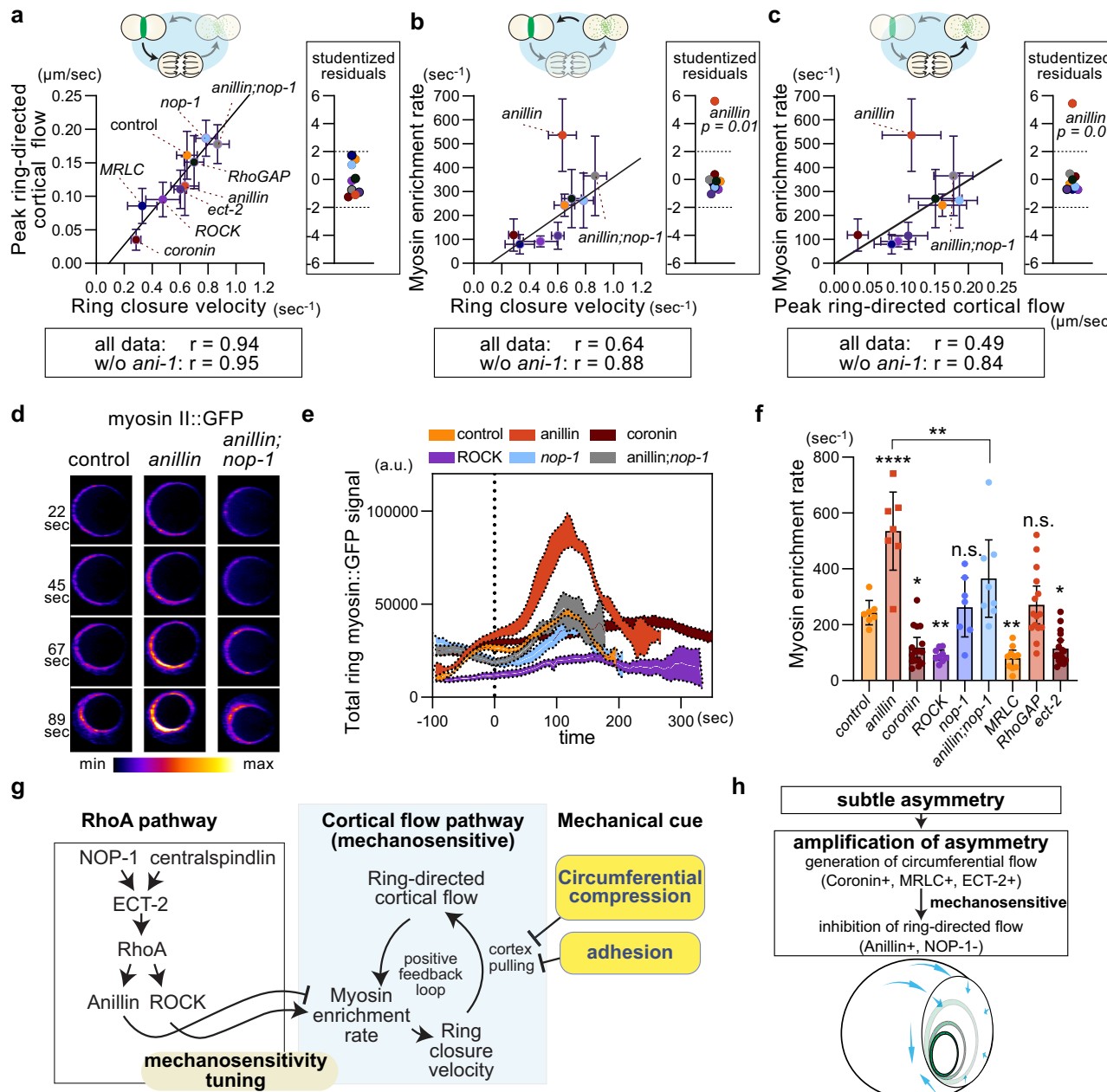

**Fig. 9 | Anillin maintains the balance between ring closure, ring-directed flow, and myosin enrichment rate. a–c** Relationship among mean ring closure velocity, mean peak ring-directed cortical flow, and mean myosin enrichment rate. Outliers are indicated by studentized residuals. **d–f** Analysis of ring myosin enrichment. The total myosin signal in the ring was plotted over time (**e**), using ring en face view images (**d**). Myosin enrichment rate is the slope of lines connecting local minima and maxima in graph e after cytokinesis onset and the mean value is shown in (**f**).

**g** Proposed model of contractile ring mechanosensation and its tuning. See texts. **h** Proposed model of asymmetric ring closure in zygotes. Error bars and bands indicate standard deviation (**a–c**), standard error of the mean (**e**), and 95% confidence intervals (**f**). Times are relative to cytokinesis onset. Scale bars, 10 μm. $p$-values were calculated by one-way ANOVA with Holm-Sidak's multiple comparison test. Source data are provided in a Source Data file.

patch formation (inhibitory arrow in Fig. 9g). Consistent with both scenarios, normal myosin enrichment rate, as well as ring mechanosensitivity, were restored by the mutation of the RhoA activator *nop-1* (Figs. 8 and 9f). Although the biochemical function of NOP-1 remains unclear, NOP-1 localizes to the cleavage furrow and upregulates the cortical localization of RhoA effectors in different axes, such as anillin, myosin, and actin[52]. This seems unlikely if NOP-1 belongs to one of the downstream axes. Furthermore, NOP-1 is essential for contractility and RhoA activation in the absence of a centralspindlin component CYK-4[52,64]. Thus, it is plausible to consider that NOP-1 is upstream of RhoA and positively regulates ring mechanosensitivity (Fig. 9g). Finally, we

observed that the peak ring eccentricity was increased in ROCK KD (Fig. 2d). Therefore, we propose that the RhoA pathway tunes ring mechanosensitivity by modulating the myosin enrichment rate (Fig. 9g). Future studies would require structure-function analysis of anillin to uncover the molecular mechanisms underlying the regulation of myosin enrichment rate within the ring.

Our study leveraged a simple model system without neighboring cells and extrinsic forces. However, our data from the two-cell stage, as well as the in vitro cell-bead adhesion assay, suggest that intercellular adhesive forces also serve as a mechanical cue, regulating unilateral cytokinesis through mechanical inhibition of ring-directed flow

(Fig. 7). Additionally, similar to zygotes, the two-cell stage ring eccentricity was reduced in anillin KD but rescued by the *nop-1* mutation (Supplementary Fig. 6). Hence, our model should be valuable in understanding the mechanosensation of the contractile ring in multicellular contexts. To achieve this goal, it is imperative to integrate other mechanosensitive mechanisms into this model, such as the mechanosensitive regulation of myosin in neighboring cells during division[65,66].

The fine-tuning of ring mechanosensitivity is likely crucial in animal development. During lumen morphogenesis, epithelial cells initially undergo symmetric cytokinesis to form a central lumen[15,67–70]. Subsequent ring closures become asymmetric towards the existing lumen, maintaining a solitary central lumen. However, in the renal cells depleted of intraflagellar transport protein IFT88, the first cytokinesis becomes abnormally asymmetric, leading to the formation of multiple lumens and polycystic kidney phenotype in mouse [15,71]. In this context, the contractile ring in the first cytokinesis should exhibit lower mechanosensitivity to prevent ectopic unilateral cytokinesis, while the contractile ring in subsequent cytokinesis should be highly mechanosensitive to be guided to the correct position. Our study provides a model and tools for a deeper understanding of 4D mechanochemical regulation of the contractile ring, which is crucial for further elucidating the role of cytokinesis in these morphogenetic events.

## Methods

### Experimental model and subject details
All *C. elegans* strains were cultured using the standard method[72]. A temperature sensitive-actin mutant *act-2(or295)* was cultured at 15 °C until the L4 larval stage and incubated at 25 °C overnight before imaging. The following transgenes were used: *cp13[nmy-2::GFP + LoxP]* (non-muscle myosin II)[73], *or1940[GFP::sas-7]* (centriole marker)[74], *itIs37* (mCherry::histone H2B), *ruIs32[pie-1p::GFP::H2B::pie-1 3'UTR + unc-119(+)]*, *knu83[cyk-1/formin::GFP]*[75], *ca725[arx-2/Arp2::TagRFP]*[76], and *ca973[GFP::pod-1/coronin]*[49]. Other than *itIs37*, all the fluorescent reporters are endogenously tagged and were maintained as homozygotes.

### RNAi
Feeding RNAi was performed at 25 °C using the standard method[77]. For control RNAi, a bacterial strain carrying an empty vector (L4440) was used. For *ani-1*/anillin and *let-502*/ROCK knockdown in $P_0$ zygotes, the L2 stage larvae were cultured on freshly prepared feeding RNAi plates on day 1. The L4 larvae were then transferred to new feeding RNAi plates on day 2 and imaged on day 3. For *let-502*/ROCK knockdown in the AB cell, we performed L4 RNAi to avoid abnormal AB spindle orientation. For *pod-1*/coronin, *mlc-4*/MRLC, and *zen-4*/MKLP, L4 larvae were cultured on feeding RNAi plates and used for imaging on the next day. For *ect-2* knockdown, young adult worms were cultured for 6hrs on the RNAi plates before imaging. For other RNAi, L1 larvae were used. All the feeding vectors with different sources[4,78,79] were sequence verified and confirmed their effectiveness based either on the reported RNAi phenotypes or the loss of signals of endogenously tagged fluorescent reporters. Sequences for RNAi are provided in Supplementary Data 1.

### Blastomere isolation
Blastomeres were isolated as described before[80], with some modification[81]. We cut the gravid adult worms in egg salt buffer and treated them with hypochlorite solution [75% Clorox (Clorox) and 2.5 N KOH] for 50 s. After washing twice with Shelton's growth medium[82], embryos were transferred to fresh Shelton's growth medium. Eggshell and permeability barrier were removed by mouth pipetting with hand-drawn microcapillary tubes (10 µL, Kimble Glass Inc.). The two-cell stage eggshell-free embryos were further pipetted to remove the cell-cell contact.

### Adhesive polystyrene bead preparation
The detailed method is described in our previous papers[4,81]. Approximately 10 mg carboxyl-modified polystyrene beads with diameters of 30 µm (Kisker Biotech GmbH & Co.), 20 µm, 10 µm, and 4.5 µm (Polysciences) were washed twice with 100 mM 2-(N-morpholino) ethanesulfonic acid (MES) buffer (pH6.5) and incubated with 1 mL MES buffer containing 10 mg 1-Ethyl-3-(3-dimethylaminopropyl) carbodiimide (EDAC) for 15 min at room temperature (22.5 °C). We washed the beads twice with phosphate-buffered saline (PBS) and incubated them with 0.5 mL PBS containing 0.05 µg Rhodamine Red-X succinimidyl ester (ThermoFisher Scientific) for 5 min. The appropriate concentration was also determined by treating the beads with a series of serially diluted Rhodamine Red-X solution[81]. The beads were washed twice with PBS and stored in PBS at 4 °C. Adhesiveness of the beads was confirmed by attaching them to the isolated blastomere using the mouth pipette. If successful, the adhesion is firmly established, and cells do not dissociate spontaneously.

### Live-imaging sample preparation
To obtain embryos, gravid adults were dissected on a coverslip, in a droplet of 10–12 µl of refractive index-matching medium (30% iodixanol diluted in egg salt buffer, supplemented with 30 µm diameter plain polystyrene beads) as described before[44]. After placing the coverslip gently onto a slide glass, three edges of coverslip were sealed with petroleum jelly (Vaseline), with one edge remaining open to the air. This method improves the success rate of imaging for inexperienced users. Inexperienced users may observe cell death due to the acute compression during the sample preparation and require training using control strains. If all the processes were performed correctly, the imaging condition does not have adverse effects on the embryonic viability, as judged by normal cell division in the next cell cycle.

### Microscopy
Intact embryos were imaged using a microscope Olympus IX83 (Olympus), equipped with a spinning-disk confocal unit CSU-W1 (Yokogawa), a scientific CMOS camera Prime 95B (Photometrics), a piezoelectric stage NANO-Z (Mad City Labs), a silicon immersion objective UPLSAPO60XS2 (NA1.3, 60X; Olympus), and a beam splitter Optosplit II (Cairn Research), which is controlled by Cellsense Dimension (Olympus). A silicone immersion oil (Z81114; refractive index: 1.406 at 23 °C; Olympus) was used as an immersion medium. Samples were illuminated by diode-pumped lasers with 488 nm and 561 nm wavelengths, and the simultaneous two color-imaging was performed with 150 ms camera exposure time, 1 µm Z-step size with a total of 31 slices per frame, 5.6-s interval, and the duration of 15 min. Isolated blastomeres were imaged using a microscope Leica DMi8 (Leica Microsystems), equipped with a spinning-disk confocal unit CSU-W1 with Borealis (Andor Technology), dual EMCCD cameras iXon Ultra 897 (Andor Technology), and an oil-immersion PL APO objective lens (NA1.4, 63X; Leica), and controlled by Metamorph (Molecular Devices). Data in Figs. 7b and 7f were imaged with 1.5 µm Z-step size and 15-s intervals. Data in Fig. 7D were imaged with 1.5 µm Z-step size and 10-s intervals, with only the half volume closer to the objective lens imaged.

### Quantification of the $P_0$ contractile ring dynamics
The obtained 4D data were deconvoluted using a constrained iterative and advanced maximum likelihood algorithm (iteration: 5), using Olympus Cellsens software (Olympus, Inc). Each 4D tiff stack file was processed using Fiji[83] (Fig. S1). The deconvoluted images were processed using Gaussian blur (sigma = 2) and an image J plug-in "attenuation correction" (opening = 3, reference = 15)[84]. A 10-µm W x 32 µm H boxed region corresponding to the contractile ring was selected and adjusted for the fluorescence intensity so that the signal would not be saturated in the next step, and rotated by 3D projection

(Brightest point, interpolation on). After selecting a plane of en face ring view, the images were segmented and quantified using an Image J plug-in Morpholib J[85]. The segmented contractile ring areas were measured for ring radius, ring centroid, and ring angle. The radius of the segmented area was estimated using ellipsoid fitting and derived by calculating the average of major and minor radii of the ellipsoid. Data were aligned relative to the time point first exceeded 10% ring closure. The ring trajectory images were obtained with an in-house image J macro using same data.

Ring closure is defined as follows:

$$R_{closure}(t) = \frac{R_0 - R_t}{R_0} \tag{1}$$

Ring eccentricity is defined as follows:

$$E(t) = \frac{Q_t}{R_0} \tag{2}$$

Leading-edge ingression is defined as follows:

$$L_{lead}(t) = \frac{R_0 + Q_t - R_t}{2R_0} \tag{3}$$

Lagging-edge ingression is defined as follows:

$$L_{lag}(t) = \frac{R_0 - Q_t - R_t}{2R_0} \tag{4}$$

### Quantification of the intact AB contractile ring dynamics

Since the AB cell undergoes rotation during its division, the precise quantification is challenging. We first selected samples undergoing unilateral cleavage roughly in parallel to the imaging plane, and corrected for cell rotation using Stackreg plug-in of Image J[86], using chromosome signal (polar bodies were deleted using a brush tool to avoid abnormal image registration, Supplementary Fig. 5a). The rotation-corrected images were then processed by the same pipeline used for the $P_0$ cell.

### 3D visualization and quantification of contractile ring dynamics in isolated blastomeres

The oblique 3D view of cells were generated using an Image J plug-in Clear Volume[87]. The analysis of the contractile ring dynamics in isolated blastomeres is challenging since the cell is rotated along different axes, and not all the planes were captured during imaging. Thus, we made en face ring view images for each time point using Clear Volume, and estimated the cell and ring contours, centroids, and diameters, by selecting more than four points along the cell and ring perimeter, respectively, using an Image J macro "Smallest Enclosing Circle[83]". The ring closure rate and eccentricity were calculated using the method described above for other cells. Measurements were performed three times per sample, and average values were used to mitigate the relatively higher error rate compared to the automatic segmentation method used for intact embryos.

### Kymographs

Kymographs in Fig. 5a were generated by stacking 1 μm H x 40 μm W rectangular regions. Kymographs in Fig. 5b–d were generated by stacking 11 μm W x 3.7 μm H rectangular regions including a myosin cluster. The timing is adjusted so that the peak myosin cluster intensity comes fourth out of a total of eight frames, using an in-house image J macro.

### Quantification of longitudinal strain rate

To measure the longitudinal strain rate in Fig. 8c, we tracked the positions of myosin foci at the lagging cell cortex. As depicted in Fig. 3c (left panel), we measured the distance between myosin foci at the top and bottom edges of the equatorial lagging cortex for 27.9 s, and then calculated the changes in their distances to derive the longitudinal strain rate.

### Quantification of myosin enrichment rate

We first measured total myosin signal in the ring using the ring en face view of myosin II/NMY-2::GFP, as shown in Supplementary Fig. 6. Selection of the contractile ring region was performed using the segmentation data described above. However, unlike the highly pre-processed data used for segmentation (e.g., attenuation corrected), we used raw imaging data without deconvolution and other processing. The only image processing performed were 3D projection and interpolation. After binarizing the segmentation data, we dilated the mask four times (mask 1) and also created another mask with four times of erosion (mask 2). In our condition, mask1 covered the area inside the outer ring perimeter, while mask 2 covered the area inside the ring (cytoplasm). The total ring myosin signal at each time point was then calculated by subtracting the signal in mask 2 from the signal in mask 1. The timing of these time series data was aligned relative to 10% ring closure. Myosin enrichment rate was calculated based on the increase in the total ring myosin signal over time. More specifically, we calculated the slopes of lines connecting local minima and maxima of the total ring myosin signal after cytokinesis onset.

### Particle image velocimetry

A detailed image preprocessing pipeline is shown in Supplementary Fig. 4. Briefly, 4D data sets of myosin II/NMY-2::GFP and GFP::SAS-7-expressing embryos were processed with Gaussian Blur (sigma = 1) and unsharp mask (radius = 1, mask = 0.6) using Fiji. Centriole signals were deleted by filling zero values. After intensity adjustment to avoid signal saturation, half volume of image stack was filled with zero values to avoid projecting cortical myosin from opposite cortical sides. The images obtained were projected using an Image J function "3D projection." We rotated 3D projected data relative to angle of cleavage at 25% closure and generated leading and lagging cortex views. Note that not always both lagging and leading cortices were clearly visible, so that in some embryos, either the leading or lagging cortex was used. Using these flattened cell cortex images, PIV was performed using MATLAB software PIV LAB (Algorithm: FFT window deformation, Interrogation area: 5.8 μm, Step size: 2.9 μm, Sub-pixel estimator: Gauss 2 × 3 point, Correlation robustness: Standard)[57]. We rejected vectors exceeding 0.25 μm/s to remove estimation errors (the setting was visually confirmed to reflect the myosin foci movement). Obtained 3D matrix data (2D x time) were used for downstream analyses. The PIV matrix data were aligned relative to 10% ring closure, centered relative to the cell center corresponding to the furrow position, and resized to contain an entire embryo using Numpy and Pandas[88,89].

### Estimation of cortical convergence

Cortical convergence was derived using the PIV vector matrix and a NumPy gradient function[88]. When we define A-P-axis and circumferential-axis cortical flow velocities in an i x j matrix as $\mathbf{u}_{i,j}$ and $\mathbf{v}_{i,j}$, respectively, $\mathbf{u}_{i+h,j}$, $\mathbf{u}_{i-h,j}$, $\mathbf{v}_{i,j+h}$, $\mathbf{v}_{i,j-h}$ can be defined as follows using Taylor series[90]:

$$\mathbf{u}_{i+h,j} = \mathbf{u}_{i,j} + h\frac{\partial \mathbf{u}_{i,j}}{\partial i} + \frac{h^2}{2!}\frac{\partial^2 \mathbf{u}_{i,j}}{\partial i^2} + \mathscr{O}\left(h^3\right) \tag{5}$$

$$\mathbf{u}_{i-h,j} = \mathbf{u}_{i,j} - h\frac{\partial \mathbf{u}_{i,j}}{\partial i} + \frac{h^2}{2!}\frac{\partial^2 \mathbf{u}_{i,j}}{\partial i^2} + \mathcal{O}\left(h^3\right) \tag{6}$$

$$\mathbf{v}_{i,j+h} = \mathbf{v}_{i,j} + h\frac{\partial \mathbf{v}_{i,j}}{\partial j} + \frac{h^2}{2!}\frac{\partial^2 \mathbf{v}_{i,j}}{\partial j^2} + \mathcal{O}\left(h^3\right) \tag{7}$$

$$\mathbf{v}_{i,j-h} = \mathbf{v}_{i,j} - h\frac{\partial \mathbf{v}_{i,j}}{\partial j} + \frac{h^2}{2!}\frac{\partial^2 \mathbf{v}_{i,j}}{\partial j^2} + \mathcal{O}\left(h^3\right) \tag{8}$$

And we will obtain the following by subtraction:

$$\mathbf{u}_{i+h,j} - \mathbf{u}_{i-h,j} = 2h\frac{\partial \mathbf{u}_{i,j}}{\partial i} + \mathcal{O}\left(h^3\right) \tag{9}$$

$$\mathbf{v}_{i,j+h} - \mathbf{v}_{i,j-h} = 2h\frac{\partial \mathbf{v}_{i,j}}{\partial j} + \mathcal{O}\left(h^3\right) \tag{10}$$

Thus, the cortical flow gradients at the interior points can be defined as follows:

$$\frac{\partial \mathbf{u}_{i,j}}{\partial i} = \frac{\mathbf{u}_{i+h,j} - \mathbf{u}_{i-h,j}}{2h} + \mathcal{O}\left(h^2\right) \approx \frac{\mathbf{u}_{i+h,j} - \mathbf{u}_{i-h,j}}{2h} \tag{11}$$

$$\frac{\partial \mathbf{v}_{i,j}}{\partial j} = \frac{\mathbf{v}_{i,j+h} - \mathbf{v}_{i,j-h}}{2h} + \mathcal{O}\left(h^2\right) \approx \frac{\mathbf{v}_{i,j+h} - \mathbf{v}_{i,j-h}}{2h} \tag{12}$$

At the boundary, first order one-sided differences were achieved as follows:

$$\frac{\partial \mathbf{u}_{i,j}}{\partial i} = \frac{\mathbf{u}_{i+h,j} - \mathbf{u}_{i,j}}{h} + \mathcal{O}(h) \approx \frac{\mathbf{u}_{i+h,j} - \mathbf{u}_{i,j}}{h} \tag{13}$$

$$\frac{\partial \mathbf{v}_{i,j}}{\partial j} = \frac{\mathbf{v}_{i,j+h} - \mathbf{v}_{i,j}}{h} + \mathcal{O}(h) \approx \frac{\mathbf{v}_{i,j+h} - \mathbf{v}_{i,j}}{h} \tag{14}$$

A-P-axis and circumferential-axis cortical convergence $\mathbf{p}_{i,j}$, and $\mathbf{q}_{i,j}$, respectively, were calculated as follows, where $a$ is the PIV step size:

$$\mathbf{p}_{i,j} = -\frac{1}{a}\frac{\partial \mathbf{u}_{i,j}}{\partial i} \approx -\frac{1}{a}\frac{\left(\mathbf{u}_{i+1,j} - \mathbf{u}_{i-1,j}\right)}{2}(at\ the\ interior\ points) \tag{15}$$

$$\mathbf{p}_{i,j} = -\frac{1}{a}\frac{\partial \mathbf{u}_{i,j}}{\partial i} \approx -\frac{1}{a}\left(\mathbf{u}_{i\pm 1,j} - \mathbf{u}_{i,j}\right)(at\ the\ boundary) \tag{16}$$

$$\mathbf{q}_{i,j} = -\frac{1}{a}\frac{\partial \mathbf{v}_{i,j}}{\partial j} \approx -\frac{1}{a}\frac{\left(\mathbf{v}_{i,j+1} - \mathbf{v}_{i,j-1}\right)}{2}(at\ the\ interior\ points) \tag{17}$$

$$\mathbf{q}_{i,j} = -\frac{1}{a}\frac{\partial \mathbf{v}_{i,j}}{\partial j} \approx -\frac{1}{a}\left(\mathbf{v}_{i,j\pm 1} - \mathbf{v}_{i,j}\right)(at\ the\ boundary) \tag{18}$$

To quantify leading and lagging cortex convergence at the equatorial region, we calculated the mean convergence within a $6 \times 3$ grid ($14.6\,\mu m$ W x $5.9\,\mu m$ H) around the center of the furrow position.

## Myosin foci tracking

Myosin foci were tracked using Manual Tracking plug-in of Image J in Figs. 3a, 7e, and 8a. In Fig. 7e, cell position was registered using image J plug-in Stackreg, to eliminate the movement of myosin foci caused by the movement of cell body.

## Units of measurement

In several figures, we presented data in arbitrary units. The degree of ring closure was shown, with a value of 1.0 indicating complete ring closure. On the other hand, eccentricity reaches 1.0 when the distance between the centroids of the initial and current ring equals the radius of the initial ring (the theoretical maximum possible ring off-centering at the end of cytokinesis). Cortical flow and flow convergence data were displayed in $\mu m/sec$ and $sec^{-1}$, respectively. However, when normalized time was used, they were shown in arbitrary units.

## Statistics and reproducibility

For multiple comparisons, one-way ANOVA with the Holm–Sidak's method was used. The comparison of two data was performed using the Welch's t-test. Error bars or bands correspond to the 95% confidence interval unless stated otherwise in figure legends. No statistical method was used to predetermine the sample size. The experiments were not randomized. The investigators were not blinded to the study. Statistical analyses were performed using either SciPy or Prism 9 (Graphpad). Symbols such as "****," "***," "**," and "*" indicate $p < 0.0001$, $p < 0.001$, $p < 0.01$, and $p < 0.05$, respectively. Sample sizes: Fig. 2a ($n = 39$), Fig. 2d ($n = 39, 19, 18, 24, 16, 9, 18$ from the left), Fig. 2e–g (the same as Fig. 2d), Figs. 3a–c, 5a–c, 6c–e, 8a, 9d (observations $n > 10$), Fig. 6g, h ($n = 9, 11$ from the left), Fig. 7c ($n = 10, 10, 13, 18$ from the left), Fig. 7e ($n = 15, 15, 25, 25, 25, 25, 24, 25$ from the left). Figure 8c ($n = 28, 16, 19, 8, 12, 11, 13, 8$ from the left), Fig. 8d ($n = 29, 16, 14, 17, 12, 9, 8$ from the left), Fig. 9a (control 27, anillin 19, coronin 16, ROCK 8, MRLC 7, nop-1 10, anillin; nop-1 14, RhoGAP 20, ect-2 10), Fig. 9b (control 7, anillin 7, coronin 16, ROCK 10, MRLC 10, nop-1 7, anillin; nop-1 8, RhoGAP 16, ect-2 18), Fig. 9c (control 7, anillin 7, coronin 7, ROCK 10, MRLC 10, nop-1 7, anillin; nop-1 8, RhoGAP 16, ect-2 18), Fig. 9e (same as Fig. 9b), Fig. 9f (control 7, anillin 7, coronin 16, ROCK 9, MRLC 10, nop-1 7, anillin; nop-1 8, RhoGAP 15, ect-2 17). Unless otherwise stated, the sample size indicates the number of embryos used.

## Reporting summary

Further information on research design is available in the Nature Portfolio Reporting Summary linked to this article.

## Data availability

The data used in this study is available in the Source Data file. Source data are provided with this paper.

## Code availability

The codes used in this study are available at https://github.com/sugi01/unilateralcytokinesis.

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

## Acknowledgements

We thank Kota Mizumoto and Don Moerman for critical reading of the manuscript. Some strains used in this study were obtained from the *Caenorhabditis* Genetics Center (funded by the NIH Office of Research Infrastructure Programs; P40 OD010440). We also thank Ronen Zaidel-Bar and Guangshuo Ou for sharing strains, Bruce Bowerman for providing valuable advice, Chris Doe for sharing lab equipment, and the Sugioka lab members for general discussions. This work was supported by the Canadian Institutes of Health Research (Project Grant; PJT-169145), Government of Canada's New Frontiers in Research Fund (NFRFE-2019-00310), and the Health Research BC (Scholar Award; SCH-2020-0406) to K.S. C.R.H. was supported by British Columbia Graduate Scholarship.

## Author contributions

C.R.H.: Formal analysis, Investigation, Writing-Original Draft, Validation, Visualization. G.S.: Formal analysis, Investigation, Validation, Visualization. W.F.: Formal analysis, Investigation. J.Z.: Formal analysis, Investigation. K.S.: Conceptualization, Formal analysis, Investigation, Methodology, Software, Validation, Writing-Original Draft, Writing-Review & Editing, Supervision, Funding acquisition.

## Competing interests

The authors declare no competing interests.
