## [Peer Review File · Nature Communications]

Contractile ring mechanosensation and its anillin-dependent tuning during early embryogenesisREVIEWER COMMENTS

Reviewer #1 (Remarks to the Author):

This manuscript describes the mechanical mechanisms controlling ingression of the contractile ring during cytokinesis, and how they control asymmetric vs. symmetric ingression. Cytokinesis occurs to separate the daughter cells as a crucial part of mitotic exit. While it is still not fully clear as to why some rings close asymmetrically vs. symmetrically, studies have shown they help direct different aspects of tissue development. The mechanisms that cause asymmetric ingression, especially in some cell types is not well-understood. The authors establish methods to quantify asymmetric ingression e.g., measuring the difference in timing between the leading vs. lagging part of the ring and velocity of closure, and correlating this with the degree of eccentricity. They knock down multiple genes that were previously implicated in mechanically regulating ingression, and measured the effect of these knockdowns to reveal those that play a role in eccentricity vs. velocity. They show that ring-directed cortical flows facilitate ingression, while compression, especially outside the ring, could locally inhibit ring-directed flows and myosin accumulation to cause asymmetric ingression, which is reinforced via mechanosensing. The authors conclude that there are two pathways contributing to ingression where a RhoA-dependent pathway controls myosin levels in the ring together with cortical flows, and intrinsic forces such as circumferential compression and external forces such as adhesion can locally reduce cortical flows to cause asymmetric ingression. The authors also determine that *ani-1* plays a role more specifically in ring mechanosensing vs. ring-directed flows. Overall, I think the study will add new knowledge to the field and provide us with quantitative methods and parameters to consider when studying ring closure/ingression. However, there are a few issues that should be addressed. Many of these issues could be addressed by text changes, however, there are a few recommended experiments that could provide more support for their models.

Major recommendations

1. The text suffers from lack of clarity and accessibility for readers that are less familiar with *C. elegans* and/or ring closure. There are many instances of assumed knowledge and over-simplistic descriptions of complex experiments and/or interpretation of outcomes. Rationale is often not provided, leaving the reader to wonder why a particular experiment is being done. For example, insufficient background is provided on *pod-1*, and it is not clear to the reader why this gene is being knocked down and included in their study. Similarly, there is no molecular description of anillin and its (core) interactors. The human homologue can bind to both actin and myosin and thus function as a crosslinker, which is highly relevant when considering its role in regulating mechanical properties of the cortex. It is also complicated by its requirement to bind to RhoA, septins and phospholipids in its C-terminus, and has been implicated both as a positive regulator of RhoA signalling (e.g., via phospholipid clustering) and in controlling RhoA turnover (e.g., via GCK-1, CCM-3 and RGA-3/4). This can make global *ani-1* knockdowns difficult to interpret, and even more so because they are never null in the context of *C. elegans* embryos. In the results, they should group genes by function (e.g., impact on the cortex) and clearly indicate what it is they are testing by knocking down that function. Reducing overall levels of F-actin or reducing overall levels of myosin vs. crosslinkers would have different rationale and predicted outcomes.

2. With this in mind, I find it odd that the authors do not knock down septins or *rho-1* (partial) or the RHO-1 regulators *ect-2* (partial) and *rga-3/4* to study this aspect of *ani-1* vs. more strictly manipulating actin and/or myosin (separation-of-function mutations would be ideal, but given the tight genetic relationship between septins and anillin, septin knockdowns could be sufficient). The only RHO-1 regulator they test is *nop-1*, where its putative function is highly speculative and poorly understood, and is redundant. Some of these other knockdowns/genetic tests are important because *ani-1* was shown in multiple publications (e.g., first shown by Maddox et al., 2007) to be required for asymmetric ingression, including the effect on myosin. The 'new' data is to correlate eccentricity with velocity in an attempt to separate ingression per se (e.g., ring-directed flows) vs. RhoA-dependent accumulation. If the authors want to claim that they have identified a mechanism whereby *ani-1*

regulates mechanosensing, then this needs to be more thoroughly investigated. For example, to describe ani-1's function in mechanosensing, they draw heavily on knowledge from Bell et al. 2020 which shows that ANI-1 could facilitate RHO-1-GTP turnover via GCK-1 and CCM-3, which localize RGA-3/4, the GAP for RHO-1, yet they did not test any of these components in their own study. The uniform 'enriched' myosin localization in the ring in ani-1 RNAi vs. control embryos supports that the transient local decreases in myosin could be lost, but again, this was previously shown. These earlier studies also revealed some interesting phenomena that the authors could consider in their own study – e.g., let-502 RNAi = slower velocity, more eccentricity vs. septin RNAi = faster velocity, less eccentricity, and they also showed that the double was symmetric vs. asymmetric. Interestingly, the authors report that eccentricity increases after let-502 knockdown, but mlc-4 decreases it, and yet do not comment on this – why do they think this occurs and how does it fit with their models? I see an opportunity to reveal something very interesting through this difference in this current study.

3. Related to the comment above, this raises the major issue is that all of the knockdowns performed in this study have been previously published in a similar context (e.g., multiple labs, also Singh et al., 2019 who revealed mechanical mechanisms regulating unilateral ingression), and the authors should incorporate some new conditions to test and add to the knowledge of the field. While I appreciate that they were able to place more emphasis on using metrics to separate pathways in this manuscript, this still reduces the novelty and potential impact of the current study. It doesn't help that in this manuscript the authors use even fewer probes to study the cortex such as those reporting for actin or active RhoA that could indicate more clearly how some of the different knockdowns impact contractility. The strength and novelty of this manuscript should be better emphasized – e.g., clarify what they are showing for the first time. The way I perceive the value in this study is in their use of metrics, and revealing how forces extrinsic to the ring could locally impact ingression.

4. Their model of adhesion (in the strictest sense) causing symmetric ingression to switch to unilateral ingression is not tested mechanistically. They show that very large (30 um – bigger than the P1 cell!) beads placed beside isolated AB cells can cause a change in eccentricity. This preliminary data is interesting, but the best effect is seen with two beads vs. one, and together these are so large, it is difficult to understand how this could asymmetrically affect the flows in their model to change eccentricity. They refer to adhesion, but no adhesion genes are tested. Since this is the only data exploring how extrinsic forces affect ingression, they should consider testing at least one potentially responsive gene to provide more evidence for their model. This is the only evidence that supports their interpretation of mechanics in the tissue context, and could be a very important, powerful part of their study that helps explain the switch in unilateral furrowing that has been observed in other contexts.

Minor recommendations and edits

Abstract

Define what the contractile ring is/does, and delete 'plays crucial roles in animal morphogenesis'. The study is about cell division, not tissue morphogenesis. They could end the abstract with speculation on the relevance of their findings in the tissue context.

Introduction

Line 40. The authors should tell the reader what a contractile ring is and its function, then tell us what unilateral ingression is and why it is relevant for morphogenesis.

Line 62 – The term 'leading edge' is C. elegans-specific, based on the localization of different ring components in the P0 zygote. The authors should clarify what this means.

Line 69-70 "knockdowns of anillin and septin disrupt the asymmetric ring closure" change to "the knockdown of anillin or septin disrupts asymmetric ring closure".

As emphasized above, the authors do not provide readers with enough information about the

molecular players in their study. Specifically, the molecular interactions and functions of anillin and key interactors (especially septins, actin and myosin) that could control different cortical properties for ring assembly and ingression. They also need to clarify how some of the *C. elegans* homologues may have different threshold requirements and/or functions compared to other homologues. For example, anillin is essential for cytokinesis in *Drosophila* S2 cells and HeLa cells, but lower thresholds support cytokinesis in the *C. elegans* P0 zygote. Readers will wonder how the authors are able to test a role for *ani-1* in asymmetric ingression if its homologue is essential in other cell types. This helps the reader understand why the authors can use global perturbation (e.g., RNAi).

Tell the reader more about *nop-1* and its redundancy as a RhoA regulator for cytokinesis. As above, the authors need to also test partial *ect-2* depletion in their study.

Lines 75-76 – “Recent studies show that cortical flow, a concerted flow of cell cortex materials at the cell surface”. Define what cell cortex materials are. This term was taken from Singh et al., but still needs to be explained.

Lines 77-78 “Therefore, we hypothesized that the contractile ring utilizes this mechanism to sense tissue mechanics” – what mechanism? How does cortical flow and compression within a single cell fit a model for sensing tissue? It isn’t clear to the reader how this will influence unilateral ingression.

Line 86 – confirming vs. confirmed

Results

It is not clear if all of the knockdowns work, and the extent of knockdown is not shown for any of the conditions. The authors should provide some evidence for this.

Line 92 – Clarify for the reader that the GFP tag fused to NMY-2 is endogenous (I assume it is the one generated by Dickinson) and if it is homo vs. heterozygous. The authors should also cite studies and/or show that filament assembly is not affected by the tag. Showing actin localization and changes in NMY-2 localization in response to depletion of regulators would support this (e.g., *let-502*, *mel-11*, *ect-2*, *rho-1*).

Lines 95-96 – Add a sentence to explain to the reader what ‘compression-dependent cell rotation is’. I thought that the eggshell still provided some of this external force, or is the eggshell also removed? Also, I didn’t think rotation was much of an issue for the one-cell zygote vs. the two-cell stage.

Line 106 – ‘degrees of leading-edge and lagging-edge advancement towards the center’. More explanation is needed to explain how this measurement was determined – why over 2RO?

Lines 122-123 – “ Δt , was determined by the delay between the leading and lagging edges reaching 10% advancement relative to the initial ring diameter”. This is not clear, perhaps just say onset of ingression?

Line 124 – typo ‘R closure’

Lines 128 – 133 – It isn’t clear why the authors divided the time lag by velocity. Their argument is that a ring that closes more slowly would have a longer time lag, but isn’t that the point, since velocity is determined by the change in distance over time? To divide the delta time by velocity (which is also determined by time) seems to put more weight on the time component. This is followed by the conclusion that both velocity and time lag control asymmetric ring closure, which seems confusing. I suspect I am missing something in their explanation.

Lines 134 – 141 – The rationale for knocking down the genes they chose is not clear and seems to be

strictly based on Singh et al. Some of the genes are crosslinkers while others regulate contractility, which could have different effects on cortical properties and mechanics. For example, what is the function of POD-1/coronin and why did the authors knock this gene down? The authors could consider grouping genes according to their function e.g., actin crosslinkers: *ani-1*, septin, alpha-actinin (and *pod-1*?) vs. actomyosin contractility: *let-502*, *ect-2*, *rho-1*, *nop-1*. Also, some of these genes can also affect anterior-posterior polarity and cortical flows that are not ring specific. How can the authors separate these from ring-specific phenotypes?

Lines 160 – 161 – Clarify what is meant by “we hypothesized that the delay is due to the feedback loop having lower activity at the lagging edge than at the leading edge”. Lower activity of what? Is this in reference to ring-directed flows? Do they mean that the feedback starts lower and stays lower in one region? Why would it be lower to begin with? The authors should distinguish between what could initiate vs. propagate asymmetry – e.g., if there is asymmetry to begin with, then it will be maintained by feedback.

Figure 3A legend – ‘Resulted myosin enrichment to the ring further facilitates ring constriction’ change to ‘Myosin enrichment in the ring facilitates constriction’. Also, more information should be provided to the reader to understand what is shown in the figure.

Figure 3. Why do the authors use normalized time on the X-axis vs. real time? Also, I assume the first dotted line is leading furrow onset?

I am struggling to understand the role of the circumferential vectors the authors call ‘convergence’ when they are imaging the lagging edge quite late in cytokinesis (e.g., the ring is >50% closed), so it is not surprising that the vectors will shift down vs. across.

For Figure 4, the different conditions need to be explained to the reader.

Also, “We found that conditions with asymmetric ring closure (control, *ani-1*; *nop-1*, and *let-502*) and symmetric ring closure (*ani-1* and *nop-1*) exhibited a reduction and no change in ring directed compression, respectively (Figure 4C).” Does this mean that *nop-1* closes symmetrically? This contradicts their measurements of eccentricity for the *nop-1* mutant in Fig. 2D.

Lines 211-212 – “Since control and *ani-1*(RNAi) displayed a similar ring closure velocity (Figure 2F), the observed cortical relaxation cannot be attributed to the ring closure rate.” This is not clear, explain what is meant by cortical relaxation.

Lines 221-222 “We hypothesized that any non-ring-directed cortical compression, like the circumferential-axis compression we observed, inhibits ring-directed cortical flow.” I am confused. Up until this point, the earlier text made it sound like the circumferential axis compression in Figure 3 was caused by constriction (which is ring-directed). I understand that they also see that contractility outside the ring can have an impact on ring-directed flows, but this sounds confusing. Importantly, the figure shows an ROI in the equatorial plane.

Figure 5 – The data needs to be more clearly explained. I understand what the authors are trying to show, but the way it is presented it is challenging to follow and it is not clear to the reader what is being shown and why. The images are not clearly explained in the legend or in the main text. The imaging locations, timing, etc is not described. For example, in Figure 5A, where is the leading vs. lagging cortex for the *act-5* mutant, and if symmetric, then show the reader where the ring is being measured. Figure 5B, what is being shown here? control zygote – stabilization of what? Figures 5C and D, where is this taken from in the embryo? What does ‘area behind the ring’ mean? Italicize gene names.

Also, they should refer to the measurements shown for the *act-5* mutant in Figure 2. It seems to lose

eccentricity, so does this support their model or not? How do they interpret this finding with respect to external compression and potential impact on the ring-directed flow?

Also in reference to Figures 5 and 6, in the main text, the authors could more clearly state that non-ring directed compression (e.g., ectopic contraction) can inhibit ring-directed flow. This concept is logical to follow – competition from different contractile events.

Figures 5 and 6 could be combined, with some of Figure 5 being moved to supplemental data.

Lines 236-237 – what is 'behind the ring'?

Line 243 – delete 'than' ('than the control)

The authors could gain more support for their model by co-depleting ani-1 in the act-5 mutant, where the 'anchoring' of actomyosin filaments mediated via ani-1 could be disrupted. Or do they mean something else?

Figure 6 – The trajectory of the centroids is hard to see as they are quite small, could the authors make these larger? Also, why measuring acceleration vs. displacement for the edge and center? No explanation is provided to the reader on the rationale for some of the methods used for measuring parameters of the ring.

Figure 7 – There is discrepancy between the text vs. what is shown in the figure. For example, two beads are shown in the figure, and this seems to cause the greatest impact on eccentricity, but in the text, there is no mention of this. The authors should speculate (in the Discussion) on why they needed two beads where each one is larger than an individual P1 cell to cause eccentricity that still isn't fully restored to control levels.

Also, for Figure 7 – What do D and P refer to? Are two beads ns vs. an intact cell or still different? Stats not shown for this comparison.

Figure 8 – typo – called Figure 7 on the figure.

Lines 292-295 – clarify what the reader is meant to infer from the pod-1 RNAi embryos which lose asymmetry in P0 but not in AB. What about mlc-4? The authors indicate that the loss of circumferential convergence could explain this, but not why this doesn't occur in AB. Further, they write 'These results suggest that the contractile ring in pod-1(RNAi) retains its mechanosensitivity but lacks the P0 mechanical cue.' I am not sure what they mean by this statement.

When referring the loss of mechanosensitivity for ani-1 RNAi – do they mean to say that this is for both P0 and AB cells? Clarify for the reader.

Figure 8 – The authors should refer back to and/or show velocity measurements vs. correlative plots (only some are shown earlier in Figure 2). The idea that myosin is enriched in ani-1 RNAi embryos yet closure occurs at the same rate/velocity demonstrates that the myosin enrichment alone may not be sufficient, which has been shown by other studies – breadth or distribution could be a more meaningful parameter, but it wasn't measured here. The authors should comment if they also measured breadth or if their calculation considers this parameter as well.

Discussion

In general, there are a few pieces of data that the authors should speculate on in their discussion, and include some of the caveats of their study.

Also, there is also an issue in how their model is presented (Rho-dependent ring vs. flow-directed ring), since cortical flows are still driven by RhoA and these are not necessarily separable. I understand what the authors are trying to say, but they should consider different terminology and/or be very clear in their explanation.

The Figure in 8G needs some modification. They show a negative arrow from ANI-1 to RhoA. This is not supported by their data, and if anything has been shown to have positive vs. negative feedback to RhoA. I realize that this is based on the Bell et al. paper where ANI-1 can feedback to RGA-3/4, but this could facilitate RhoA turnover to make more RhoA-GTP. The only evidence they have for this arrow in this study is based on the higher/more symmetric levels of myosin in the ring, but this doesn't mean there is more active myosin in the system (see earlier comments). One way to deal with this contradiction is to just remove the blocked arrow from the figure. They can still have ANI-1 playing a central role in their mechanosensing model in H, where it makes more sense in having more localized effects on myosin vs. global.

Lines 373-374 - "this variability suggests that the mechanosensitivity of contractile ring can be tuned differently during development, enabling tissue-specific morphogenic cytokinesis." What is morphogenic cytokinesis?

Reviewer #2 (Remarks to the Author):

The manuscript by Hsu et al. explores the mechanism of mechanosensation in the contractile ring. As a model system, the authors use the asymmetrically closing ring of the early *C. elegans* zygote. Asymmetric ring closure has been observed in a number of metazoan cells but the underlying mechanisms are incompletely understood. Previous studies implicated Anillin and other cortical proteins in asymmetric ring closure in *C. elegans*. A positive feedback model during ring constriction that relies on cortical flows and Myosin enrichment in the contractile ring has also been proposed previously. Here, Hsu et al., extend this model by carefully measuring ring closure dynamics in wild type and mutant worms. The authors developed a new live cell imaging method to measure ring dynamics, furrow ingression and cortical flows in both wild type and mutant situations. The authors propose that asymmetric ring closure is regulated by a mechanical feedback loop whereby local mechanical cues suppress ring-directed cortical Myosin flows which reduce compression-dependent Myosin enrichment in the contractile ring.

The manuscript convinces with detailed measurements and elegant genetic manipulations. The authors also use beads to manipulate non-ring directed cortical flows. The underlying questions and findings are of considerable relevance and this study could make a significant contribution to the field. However, in its current form, the manuscript is difficult to read mostly because the authors use several mechanical concepts that are either poorly defined/explained or used interchangeably. For instance, the connection between cortical flows, convergence and compression is not intuitively clear and the manuscript uses some terms very suddenly without much introduction or explanation. Maybe a supplemental figure that summarizes these concepts in graphical form would help interpret the data better. Also, I find it difficult to see how these concepts are related to asymmetric ring positioning because they appear to apply more generally to furrow formation and ingression dynamics. Addressing the issues below might help with this general comment.

Critiques:

(1) Line 75-78: Recent studies show that cortical flow, a concerted flow of cell cortex materials at the cell surface, facilitates contractile ring assembly through cortical compression³⁷⁻³⁹. Therefore, we hypothesized that the contractile ring utilizes this mechanism to sense tissue mechanics.

The concept of "cortical compression" should be briefly introduced here. Also, the subsequent

hypothesis needs to be explained in more detail. What type of tissue mechanics do the authors envision? This is particularly relevant as the model system, the P0 zygote is not surrounded by any neighboring cells.

(2) How is the parameter space in Figure 2F defined?

(3) The cartoon in Figure 3A is helpful but it would be more informative if it reflected the actual asymmetrically ingressing furrow situation.

(4) Line 168: Given that the contractility gradient continues to exist during ring assembly and closure, the ring should consistently pull the cell cortex from the polar region.

What is the evidence for this statement?

(5) The connection between cortical flow and axis convergence is not well explained in the manuscript. It makes it difficult to conclusively interpret the results shown in Figure 3 and 4. For instance, the statement from line 189 "These results suggest that circumferential-axis cortical flow negatively correlates with ring-directed cortical flow" is not really supported by the data.

(6) Line 197: "After the onset of furrow ingression, convergence at the surface represents the sum of cortical compression and ingression."

What data or previous findings support this statement?

(7) Line 229: The concept of "relaxation-induced" ring-directed cortical flow needs to be introduced.

(8) It is not clear why the bead experiment was performed in the AB cell. The mechanisms for asymmetric furrowing could be different in this cell type. To make the manuscript flow more coherently, it would be beneficial to provide bead data from the P0 zygote to correlate these experimental data with the other findings of the manuscript.

(9) Line 344: In our model, intracellular and intercellular mechanical cues locally limit ring-directed cell cortex pulling at the lagging cell cortex. This statement is unclear as the main model used here, the P0 zygote, has no cell neighbors. Thus, it is not clear what is meant by intercellular mechanical cues.

(10) The manuscript (and figures) should more carefully distinguish between the leading and lagging edge. For instance, in Figure 4, it is unclear whether compression data are shown for the leading or lagging furrow.

(11) How do the authors define "cytokinesis onset"? This is relevant to correctly understand the timing of events.

(12) What is the mechanism for the circumferential-axis directed flow?

(13) I would encourage the authors to provide a graphical model (similar to Figures 3A, 4F, 5E) at the end of the manuscript. That would also allow to include the effect of the different mutants more clearly.

(14) The y-axis in Figure 5C contains a typo.

Reviewer #3 (Remarks to the Author):

Rou Hsu and coworkers investigate the dynamics of cytokinesis in the early worm embryo. These dynamics have been studied intensively but this latest work uses higher temporal resolution and an improved imaging setup. Furthermore, the analysis focuses on how “mechanical cues” influence the contractile ring. The work itself along with the presentation are very well done with clear main figures and supplemental panels that nicely outline the workflows used. The first part of the manuscript makes some interesting and important observations, that cortical flow asymmetry accompanies ring eccentricity and that transverse flow occurs on the lagging cortex. They argue that these transverse flows lead to “cortical compression” that is responsible for the delay that leads to ring eccentricity but that this occurs without inhibiting ring constriction, possibly because of compensation. As part of a test of this model, they use beads to perturb cortical flows and induce eccentricity in a system that normally lacks it. Interestingly, they argue that Anillin promotes ring eccentricity by repressing cortical myosin activity. Overall this is a well executed and important study that makes important advances in our understanding of the relationships of cortical flows, cortical mechanics, and cytokinetic dynamics. I recommend publication with the following additional considerations.

- * The “linear part” in figure S1D concerns me. Since the data are highly nonlinear it seems like it would be challenging to find the appropriate section of data to apply the linear fit. I’m unsure how much this decision would affect the analyses derived from these data but it certainly seems like it could be a very significant effect.
- * Line 160 - it would be helpful if the authors explained what they mean by “the feedback loop having lower activity” in more detail.
- * Some of the graphs apparently plot normalized values (which would lack units) but are labeled as “AU”. It would be more helpful to provide some information about the normalization.
- * Figure 8 is mislabeled as Figure 7

We would like to thank all the reviewers for providing valuable feedback on our manuscript. We are pleased to see that all the reviewers agreed on the significance of this study as summarized below (original comment is colored in blue):

“Overall, I think the study will add new knowledge to the field and provide us with quantitative methods and parameters to consider when studying ring closure/ingression. (Reviewer 1)”, “The manuscript convinces with detailed measurements and elegant genetic manipulations... The underlying questions and findings are of considerable relevance and this study could make a significant contribution to the field. (Reviewer 2)”, “Overall this is a well executed and important study that makes important advances in our understanding of the relationships of cortical flows, cortical mechanics, and cytotkinetic dynamics. I recommend publication with the following additional considerations. (Reviewer 3).”

During revision, the preprint version of our manuscript was also highlighted in the *Molecular Biology of the Cell*, suggesting general interest of this study in the Cell biology field (<https://www.molbiolcell.org/doi/full/10.1091/mbc.P23-05-0016>).

We acknowledge that the original manuscript was not immediately accessible to a wide readership. In fact, it has been challenging for us to explain the 4D observation in a straightforward way, and we appreciate the insights of the reviewers. We have rewritten most of the manuscript to 1) improve readability, 2) remove unnecessary speculations, 3) clearly highlight new discoveries, and 4) provide more explanation of the 4D imaging data. Furthermore, we have performed key experiments as outlined below:

1. **New data:** Figure 2 and S3 include data on additional actomyosin regulators.
2. **New data:** Figure 3 and Movie 2, 4-5 were newly added to visualize the 3D movement of myosin foci.
3. **New data:** Figure 4 includes an analysis of ring-directed flow in zen-4/MKLP knockdown.
4. **New data:** Figure 8C shows the cortical compression rate, which was directly measured through deformation of the cell cortex.
5. **New data:** Figure 9E shows the temporal changes in the total ring myosin signal.
6. **New data:** Figure 9 includes data on additional actomyosin regulators.
7. **New data:** Figure S2 shows the robustness of the cytokinesis onset estimation method.
8. **Figure revision:** Figure 1 now introduces the known pathways of ring assembly (panel A) and highlights the innovative part of our 4D analysis (panel B).
9. **Figure revision:** Figure 2 now introduces the different actomyosin regulators analyzed in this study (panel B).
10. **Figure revision:** More schematics were added to better explain the data and model (Fig. 4C-D, G, 5A-E, 6G-I, 8E, 9A-C, 9H).
11. **Figure revision:** Figure S6 was added to explain the myosin quantification method.
12. **Figure revision:** The order of figures was changed to improve logical flow.
13. **Rewriting:** We have rewritten the Introduction section by improving its logical flow and providing more detailed elaboration on our research questions.
14. **Rewriting:** We have rewritten the Results section by improving its logical flow and providing detailed explanation for functions of actomyosin regulator, physical/mathematical terms, motivation of doing data-driven hypothesis formulation, and interpretation of data.
15. **Rewriting:** We have rewritten the Discussion section which now includes a model of the symmetry-breaking event (related to Fig. 9H). Also, it highlights the limitation and future goals of the study.
16. **Rewriting:** Method section now includes the validation method of RNAi.

Below please refer to the following point-by-point response.

Reviewer #1 (Remarks to the Author):

This manuscript describes the mechanical mechanisms controlling ingression of the contractile ring during cytokinesis, and how they control asymmetric vs. symmetric ingression. Cytokinesis occurs to separate the daughter cells as a crucial part of mitotic exit. While it is still not fully clear as to why some rings close asymmetrically vs. symmetrically, studies have shown they help direct different aspects of tissue development. The mechanisms that cause asymmetric ingression, especially in some cell types is not well-understood. The authors establish methods to quantify asymmetric ingression e.g., measuring the difference in timing between the leading vs. lagging part of the ring and velocity of closure, and correlating this with the degree of eccentricity. They knock down multiple genes that were previously implicated in mechanically regulating ingression, and measured the effect of these knockdowns to reveal those that play a role in eccentricity vs. velocity. They show that ring-directed cortical flows facilitate ingression, while compression, especially outside the ring, could locally inhibit ring-directed flows and myosin accumulation to cause asymmetric ingression, which is reinforced via mechanosensing. The authors conclude that there are two pathways contributing to ingression where a RhoA-dependent pathway controls myosin levels in the ring together with cortical flows, and intrinsic forces such as circumferential compression and external forces such as adhesion can locally reduce cortical flows to cause asymmetric ingression. The authors also determine that *ani-1* plays a role more specifically in ring mechanosensing vs. ring-directed flows. Overall, I think the study will add new knowledge to the field and provide us with quantitative methods and parameters to consider when studying ring closure/ingression. However, there are a few issues that should be addressed. Many of these issues could be addressed by text changes, however, there are a few recommended experiments that could provide more support for their models.

Firstly, we would like to thank the reviewer for the valuable feedback on our manuscript. This project has presented us significant challenges in conveying its complexity, and we appreciate the opportunity to improve our manuscript. After a thorough review of the Reviewer 1's comments, we have identified some misunderstandings about the novelty of our study and potential overinterpretation of previous findings, which may have adversely influenced the Reviewer 1's assessment of our paper. As the reviewer mentioned, they may be resolved after improving the clarity of our texts. However, we would like to clarify important points before proceeding with our point-by-point response.

Point 1: Previous understanding of mechanical regulation of cytokinesis

Mechanical regulation of ingression is described in the context of adhesion-dependent asymmetric ring closure in *Drosophila* epithelia (PMID: 23410938). However, this model does not consider the molecular mechanism of contractile ring assembly. In *C. elegans*, external artificial compressive forces (i.e., compressing embryos using agarose) can enhance asymmetric ring closure, but this phenomenon is demonstrated only in wild-type cells (PMID: 31519810). The cause of this enhancement was not addressed in their paper (because it was not the primary focus), but it has been demonstrated that compression induces asymmetric closure due to spindle off-centering (PMID: 14051860). The same study identified genes required for the successful completion of cytokinesis under compression (mechanostability). They conducted a small-scale RNAi screen targeting 14 genes, including 6 select actomyosin regulators, 2 polarity regulators, and 4 microtubule regulators. All the tested genes exhibited defects in cellular response to compression, with 13 out of 14 genes showing defects in cytokinesis under compression. This fascinating research uncovered the mechanisms underlying the resilience of cytokinesis, but it did not imply that these genes mechanically regulate ingression (the paper did not claim so, which is reasonable considering that almost all the tested genes showed defects). Additionally, in *C. elegans*, anillin and septin were known to regulate asymmetric ring closure (PMID: 17488632). Nevertheless, whether anillin and septin regulate the mechanical process or not remains unknown. While ring-directed cortical compression is proposed to promote ring assembly (PMID: 27719759, 29963981), this mechanism has not been studied in the context of

asymmetric ring closure. Therefore, the mechanical regulation of ingression is poorly understood before this study, and our research is the first to demonstrate the mechanism of contractile ring mechanosensation.

Point 2: Our primary objective and achievements in this study

Our primary objective is to understand the underlying cause of asymmetric ring closure, rather than focusing solely on identifying genes controlling eccentricity or velocity. We have discovered 1) the mechanism of contractile ring mechanosensation and 2) anillin-dependent tuning of mechanosensitivity. In the revised manuscript, we emphasize the importance of data-driven hypothesis formulation, illustrated through the example of morphogenetic degeneracy as described below.

Page 8, line 155:

“Next, we analyzed the potential causes of unilateral cytokinesis defects in mutants or RNAi knockdown of actomyosin regulators (a list of genes is in Figures 2B and S3A). A previous study showed that it is difficult to intuitively estimate the roles of actomyosin regulators in cortical dynamics from their biochemical function⁴⁵. For example, actomyosin components in completely different pathways can influence the common attributes in cortical dynamics, a phenomenon referred to as morphogenetic degeneracies. Therefore, we prioritized data-driven hypothesis formulation by comparing the obtained parameters throughout this study.”

Major recommendations

1. The text suffers from lack of clarity and accessibility for readers that are less familiar with *C. elegans* and/or ring closure. There are many instances of assumed knowledge and over-simplistic descriptions of complex experiments and/or interpretation of outcomes. Rationale is often not provided, leaving the reader to wonder why a particular experiment is being done. For example, insufficient background is provided on *pod-1*, and it is not clear to the reader why this gene is being knocked down and included in their study. Similarly, there is no molecular description of anillin and its (core) interactors. The human homologue can bind to both actin and myosin and thus function as a crosslinker, which is highly relevant when considering its role in regulating mechanical properties of the cortex. It is also complicated by its requirement to bind to RhoA, septins and phospholipids in its C-terminus, and has been implicated both as a positive regulator of RhoA signalling (e.g., via phospholipid clustering) and in controlling RhoA turnover (e.g., via GCK-1, CCM-3 and RGA-3/4). This can make global *ani-1* knockdowns difficult to interpret, and even more so because they are never null in the context of *C. elegans* embryos. In the results, they should group genes by function (e.g., impact on the cortex) and clearly indicate what it is they are testing by knocking down that function. Reducing overall levels of F-actin or reducing overall levels of myosin vs. crosslinkers would have different rationale and predicted outcomes.

We acknowledge that the original manuscript was not readily accessible to a wide readership. As the reviewer mentioned regarding anillin, it is difficult to estimate the role of actomyosin regulators in cortical dynamics solely based on their biochemical functions. This is the reason why we conducted data-driven hypothesis formulation through a comparative parameter analysis as described in our previous comment. In the revised manuscript, we have clarified the suggested points as follows.

1. We now include more detailed description of actomyosin regulators

Page 8, Line 163:

“First, we analyzed upstream of RhoA, such as RhoA activators and inhibitors. We found that ECT-2/RhoGEF is required for asymmetric ring closure (Figure 2C-D). Conversely, RhoGAP and a

worm specific RhoA activator NOP-1 are dispensable for unilateral cytokinesis (Figures 2D and S3B). Second, we analyzed RhoA effectors. Knockdown of ROCK increased peak eccentricity (Figures 2C-D). On the other hand, knockdown of anillin and septin resulted in symmetric ring closure (Figures 2C-D and S3B). Additionally, we found that knockdown of myosin regulatory light chain kinase/MRLC and gain-of-function mutation of actin gene *act-2*⁴⁶ reduced ring eccentricity, whereas formin knockdown did not (Figures 2D and S3B). Finally, we analyzed a pathway of branched actin network formation, which is known to inhibit myosin-dependent actin-contraction^{47,48}. Arp2 is a component of the Arp2/3 complex, which regulates branched nucleation of actin from the existing filaments⁴⁹. On the other hand, Coronin removes Arp2/3 from actin, promoting actin debranching⁵⁰⁻⁵². Although Arp2 knockdown exhibited a normal level of eccentricity, we found that Coronin knockdown resulted in symmetric ring closure. Thus, our analysis confirmed previously reported unilateral cytokinesis phenotypes for anillin⁹, septin⁹, ROCK⁹, NOP-1⁵³, and Arp2⁵⁴, with providing more quantitative information, and identified novel roles of ECT-2, Coronin, actin, and MRLC in unilateral cytokinesis.”

Page 9, Line 189:

“Anillin is a scaffolding protein that interacts with RhoA, actin, myosin, and septin during cytokinesis, facilitating contractile ring assembly ^{56,57}.”

2. **We now include schematics of molecular pathways regulating ring assembly in Figure 1A and 2B.**
3. **We summarized what is known and the new discoveries regarding the asymmetric ring closure phenotype.**

(Please note that this is a minor achievement of this paper. Our main achievement is the identification of the mechanism underlying the contractile ring mechanosensation. Additionally, we identified that anillin tunes contractile ring mechanosensitivity. This latter aspect will benefit from a more detailed analysis, as we described in the revised text. However, we have unambiguously demonstrated that the contractile ring mechanosensation mechanism.)

Page 9, Line 177:

“Thus, our analysis confirmed previously reported unilateral cytokinesis phenotypes for anillin⁹, septin⁹, ROCK⁹, NOP-1⁵⁴, and Arp2⁵⁵, with providing more quantitative information, and identified novel roles of ECT-2, Coronin, actin, and MRLC in unilateral cytokinesis.”

2. With this in mind, I find it odd that the authors do not knock down septins or rho-1 (partial) or the RHO-1 regulators ect-2 (partial) and rga-3/4 to study this aspect of ani-1 vs. more strictly manipulating actin and/or myosin (separation-of-function mutations would be ideal, but given the tight genetic relationship between septins and anillin, septin knockdowns could be sufficient).

In the revised manuscript, we have included septin, ect-2, rga-3/RhoGAP, cyk-1/formin, and arx-2/Arp2. Our primary focus in this section (related to Figure 2) is on data-driven hypothesis formulation aimed at identifying the cause of asymmetric ring closure due to the concerns regarding morphogenetic degeneracy mentioned earlier. Hypothesis-driven analysis based on the biochemical function of molecules could be prone to misguidance. Targeted molecular analysis such as structure-function analysis and the use of separation-of-function mutants become useful after identifying the key cortical dynamics. For example, we have identified that the normal level of myosin enrichment rate is important for mechanosensitivity of the contractile ring and is regulated by anillin. Based on these results, we can perform structure-function analysis of anillin focusing on this key parameter (this is our future project).

The only RHO-1 regulator they test is *nop-1*, where its putative function is highly speculative and poorly understood, and is redundant.

We have clarified the role of *nop-1* as follows in the discussion section:

Page 21, Line 458:

“Although the biochemical function of NOP-1 remains unclear, NOP-1 localizes to the cleavage furrow and upregulates the cortical localization of RhoA effectors in different axes, such as anillin, myosin, and actin⁷⁴. This seems unlikely if NOP-1 belongs to one of the downstream axes. Furthermore, NOP-1 is essential for contractility and RhoA activation in the absence of a centralspindlin component CYK-4^{54,65}. Thus, it is plausible to consider that NOP-1 is upstream of RhoA and positively regulates ring mechanosensitivity (Figure 9G).”

In addition to that, we have added *ect-2* and *rga-3*/RhoGAP knockdown in the revised manuscript (Figure 2, 8, 9).

Some of these other knockdowns/genetic tests are important because *ani-1* was shown in multiple publications (e.g., first shown by Maddox et al., 2007) to be required for asymmetric ingression, including the effect on myosin.

As mentioned above, we have summarized what is known and what is new discoveries regarding asymmetric ring closure phenotype in the revised manuscript.

The 'new' data is to correlate eccentricity with velocity in an attempt to separate ingression per se (e.g., ring-directed flows) vs. RhoA-dependent accumulation.

Here, we summarized what is new in this manuscript. The above reviewer's assessment mostly based on Figure 2 and unfortunately missed discoveries related to Fig3-9.

1. We have performed a 4D integrative analysis of cortical dynamics and ring closure in the context of unilateral cytokinesis. We have developed tools required for integrative analysis of different parameters using the same 4D datasets.
2. We have identified new genes required for asymmetric ring closure, as mentioned above.
3. We have performed data-driven hypothesis formulation and identified the cause of asymmetric ring closure, which include ring closure speed and time lag.
4. We have first shown the cortical flow in the division plane, revealing its role.
5. We have first identified circumferential flow.
6. We have first demonstrated that cortical compression inhibits ring-directed flow.
7. We have first demonstrated that cortical compression inhibits ring ingression.
8. We have first demonstrated that lagging edge furrowing delay is due to mechanical inhibition of ring-directed flow.
9. We have first demonstrated that anillin is required for the ring mechanosensation.
10. We have first demonstrated that the positive feedback loop in ring assembly is required for mechanosensation.

If the authors want to claim that they have identified a mechanism whereby *ani-1* regulates mechanosensing, then this needs to be more thoroughly investigated. For example, to describe *ani-1*'s function in mechanosensing, they draw heavily on knowledge from Bell et al. 2020 which shows that ANI-1 could facilitate RHO-1-GTP turnover via GCK-1 and CCM-3, which localize RGA-3/4, the GAP for RHO-1, yet they did not test any of these components in their own study. The uniform

'enriched' myosin localization in the ring in ani-1 RNAi vs. control embryos supports that the transient local decreases in myosin could be lost, but again, this was previously shown.

The order of figures in the original manuscript was confusing and corrected in the revised manuscript. We have identified the mechanism underlying contractile ring mechanosensation based on the evidence in Figure 3-7, and 8C. Subsequently, we demonstrated that ani-1 is required for mechanosensitivity tuning in Figure 8. Additionally, we demonstrated the role of anillin in maintaining a positive-feedback loop in ring assembly in Figure 9. This comment from the reviewer pertains to the data presented in Figure 9. We acknowledge that this last part would benefit from more detailed analysis, as described in the revised text. However, we unambiguously showed that contractile ring mechanosensation mechanism involving cortical flow.

The loss of asymmetric myosin enrichment in the ring in ani-1/anillin(RNAi) was previously reported in Maddox et al., 2007, and our paper is not claiming that we rediscovered this well-known phenomenon. It's important to note that the simple loss of asymmetry should still preserve the total ring myosin. Previous studies did not claim an increase in myosin enrichment. Therefore, the statement "this was previously shown" is unfortunately inaccurate.

Furthermore, what we have demonstrated is the myosin enrichment rate, which represents the speed of myosin enrichment. In ani-1/anillin (RNAi), myosin enrichment rate is no more limited by the ring-directed flow and ring closure velocity (Fig. 9A-C), unlike in other conditions. This relationship holds true based on correlational analysis, regardless of the underlying molecular mechanism.

These earlier studies also revealed some interesting phenomena that the authors could consider in their own study – e.g., let-502 RNAi = slower velocity, more eccentricity vs. septin RNAi = faster velocity, less eccentricity, and they also showed that the double was symmetric vs. asymmetric. Interestingly, the authors report that eccentricity increases after let-502 knockdown, but mlc-4 decreases it, and yet do not comment on this – why do they think this occurs and how does it fit with their models? I see an opportunity to reveal something very interesting through this difference in this current study.

As shown in Figure S3D, the peak eccentricity of all the actomyosin regulators tested can be explained by the product of ring velocity and time lag (= normalized time lag).

In Figures 3-8, we conducted cell biological analyses to determine the cause of time lag generation. We demonstrated that cortical compression generates time lag due to the suppression of ring-directed flow (Figure 5-7). In zygotes, a source of cortical compression is circumferential flow (Figure 3 and 8C). While mlc-4/MRLC knockdown resulted in the loss of circumferential compression, ROCK knockdown exhibited a normal level of circumferential compression (Fig. 8C). This explains the differences in their phenotypes.

3. Related to the comment above, this raises the major issue is that all of the knockdowns performed in this study have been previously published in a similar context (e.g., multiple labs, also Singh et al., 2019 who revealed mechanical mechanisms regulating unilateral ingression), and the authors should incorporate some new conditions to test and add to the knowledge of the field. While I appreciate that they were able to place more emphasis on using metrics to separate pathways in this manuscript, this still reduces the novelty and potential impact of the current study. It doesn't help that in this manuscript the authors use even fewer probes to study the cortex such as those reporting for actin or active RhoA that could indicate more clearly how some of the different knockdowns impact contractility. The strength and novelty of this manuscript should be better emphasized – e.g., clarify what they are

showing for the first time. The way I perceive the value in this study is in their use of metrics, and revealing how forces extrinsic to the ring could locally impact ingression.

Shing et al., 2019 demonstrated that compressing cells using external forces increases ring eccentricity in wildtype cells. They also identified genes required for successful cytokinesis completion under compression. However, their study did not specifically analyze these genes in the context of asymmetric ring closure. Therefore, the statement “(they) revealed mechanical mechanisms regulating unilateral ingression” is unfortunately a misinterpretation of their interesting paper. In our study, we identified new genes required for asymmetric ring closure, even though it was not our primary focus. Consequently, the statement “all of the knockdowns performed in this study have been previously published in a similar context” does not actually capture what we have presented. We acknowledge that these concerns arise partly from our original writing, and we fully agree with the reviewer’s remark that “The strength and novelty of this manuscript should be better emphasized”. Thus, we have rewritten our manuscript, as summarized on the first page of this response.

4. Their model of adhesion (in the strictest sense) causing symmetric ingression to switch to unilateral ingression is not tested mechanistically. They show that very large (30 μm – bigger than the P1 cell!) beads placed beside isolated AB cells can cause a change in eccentricity. This preliminary data is interesting, but the best effect is seen with two beads vs. one, and together these are so large, it is difficult to understand how this could asymmetrically affect the flows in their model to change eccentricity. They refer to adhesion, but no adhesion genes are tested. Since this is the only data exploring how extrinsic forces affect ingression, they should consider testing at least one potentially responsive gene to provide more evidence for their model. This is the only evidence that supports their interpretation of mechanics in the tissue context, and could be a very important, powerful part of their study that helps explain the switch in unilateral furrowing that has been observed in other contexts.

Here, we are referring to physical adhesion forces and not specific adhesion molecules. While the diameter of the P1 cell is smaller than the 30 μm bead, the contact areas shown in Figure 7A and Figure 7F are comparable. How this affects the flow is shown in Figure 7E. We have addressed these observations more clearly in the revised manuscript. Our future studies will further elucidate the regulation of cytokinesis in multicellular contexts. This manuscript represents a necessary step forward in achieve this overarching goal.

Page 16, Line 344:

“A single bead attachment resulted in a slight increase in the average peak eccentricity, although this change was not statistically significant (Figure 7C, $p = 0.11$). This result may be attributed to the smaller contact area compared to *in vivo* conditions (Figures 7A and 7D) and the normal ring-directed flow at the cell cortex distal to the bead (Figure 7E). Notably, the attachment of the two 30 μm beads increased peak eccentricity, and about 50% of the cells exhibited a level of ring eccentricity comparable to that of the intact AB cell *in vivo* (Figure 7C, F–H). Thus, we conclude that mechanical obstruction of ring-directed cortical flow is sufficient to induce asymmetric ring closure.”

Minor recommendations and edits

Abstract

Define what the contractile ring is/does, and delete ‘plays crucial roles in animal morphogenesis’. The study is about cell division, not tissue morphogenesis. They could end the abstract with speculation on the relevance of their findings in the tissue context.

Thank you for the comment. This study primarily concerns the regulation of contractile ring function with respect to cellular mechanics, which remains an open question in the developmental control of

cell division. While we build upon decades of rigorous cytokinesis studies in this field, our central focus lies in understanding its asymmetric regulation rather than the mechanism of cytokinesis itself. The importance of our study in the context of developmental biology was also acknowledged in a recent short review published in *Molecular Biology of the Cell*, which highlighted this manuscript (<https://www.molbiolcell.org/doi/full/10.1091/mbc.P23-05-0016>).

We have modified the abstract with addressing the concerns as follows:

Abstract: Page 2, Line 22:

“Cytokinesis plays crucial roles in morphogenesis. Previous studies have examined how tissue mechanics influences the position and closure direction of the contractile ring. However, the mechanisms by which the ring senses tissue mechanics remain largely elusive. Here, we show the mechanism of contractile ring mechanosensation and its tuning during asymmetric ring closure of *Caenorhabditis elegans* embryos. Integrative analysis of ring closure and cell cortex dynamics revealed that mechanical suppression of the ring-directed cortical flow is associated with asymmetric ring closure. Consistently, artificial obstruction of ring-directed cortical flow induces asymmetric ring closure in otherwise symmetrically dividing cells. Anillin is vital for mechanosensation. Our genetic analysis suggests that the positive feedback loop among ring-directed cortical flow, myosin enrichment, and ring constriction constitutes a mechanosensitive pathway driving asymmetric ring closure. These findings and developed tools should advance the 4D mechanobiology of cytokinesis in more complex tissues.”

Introduction

Line 40. The authors should tell the reader what a contractile ring is and its function, then tell us what unilateral ingression is and why it is relevant for morphogenesis.

Our main question is the mechanical regulation of the contractile ring, and this topic should be addressed in the first paragraph. We have modified the introduction as follows:

Page 3, Line 39:

“The cytokinetic contractile ring physically partitions the dividing cell during cell division, but it also plays pivotal roles in morphogenesis by regulating its position and function along the body axis¹⁻³. This is evident in processes such as asymmetric cell division and epithelial morphogenesis, where the position and closure of the ring are asymmetrically regulated to control the size, shape, and arrangements of daughter cells. Although previous studies have suggested that both intracellular and extracellular mechanics influence this “morphogenetic cytokinesis”⁴⁻⁶, the mechanism by which the contractile ring senses mechanical cues and modulates its function in response remain largely unexplored.”

Line 62 – The term ‘leading edge’ is *C. elegans*-specific, based on the localization of different ring components in the P0 zygote. The authors should clarify what this means.

The term “leading edge” is not *C. elegans* specific but is widely recognized in the context of cell migration (it can be found in any cell biology textbook), and it simply refers to the edge that is leading compared to other parts. We have defined it in the revised introduction as follows:

Page 4, Line 61:

“Consequently, the off-centering of the spindle along the transverse axis relative to the pole-to-pole axis induces furrowing at one side of the cell cortex (hereafter we call this site as the leading edge) ³²⁻³⁴”

Line 69-70 “knockdowns of anillin and septin disrupt the asymmetric ring closure” change to “the knockdown of anillin or septin disrupts asymmetric ring closure”.

Due to major rewriting, we have removed this sentence but eliminated the term “knockdowns” from the manuscript.

As emphasized above, the authors do not provide readers with enough information about the molecular players in their study. Specifically, the molecular interactions and functions of anillin and key interactors (especially septins, actin and myosin) that could control different cortical properties for ring assembly and ingression. They also need to clarify how some of the *C. elegans* homologues may have different threshold requirements and/or functions compared to other homologues. For example, anillin is essential for cytokinesis in *Drosophila* S2 cells and HeLa cells, but lower thresholds support cytokinesis in the *C. elegans* P0 zygote. Readers will wonder how the authors are able to test a role for *ani-1* in asymmetric ingression if its homologue is essential in other cell types. This helps the reader understand why the authors can use global perturbation (e.g., RNAi).

We have added more detailed explanation of each component in the revised manuscript, as described in our previous comment. The latter part of the comment regarding the threshold appears to be unnecessary in the context of our paper as our aim is not to identify genes regulating asymmetric ring closure. We have added a sentence regarding RNAi as follows:

Page 8, Line 162:

“Note that our knockdown condition is mild for some genes, allowing cytokinesis completion.”

Tell the reader more about *nop-1* and its redundancy as a RhoA regulator for cytokinesis. As above, the authors need to also test partial *ect-2* depletion in their study.

We have performed a partial *ect-2* knockdown, as shown in Figure 2C-F, 8C-D, and 9A-C, F in the revised manuscript. It abolished circumferential compression, as shown in Fig.8C, and resulted in symmetric closure similar to MRLC and coronin knockdown. We have modified the discussion section based on these results and added a description of *nop-1* as follows:

Page 20, Line 446:

“Our analysis suggests that components of the RhoA pathway tune contractile ring mechanosensitivity (Figure 9G; left box). We demonstrate impaired mechanosensitivity in anillin KD embryos, in which myosin was too rapidly enriched in the contractile ring (Figure 9). This observation is somewhat counterintuitive, considering the role of anillin as a scaffolding protein for actomyosin components and its involvement in the formation of large cortical patches, containing myosin^{61,62}. Recent research has shown that Anillin-binding proteins GCK-1 and CCM-3 negatively regulate RhoA activity by promoting RhoGAP cortical localization⁶³. Therefore, one possibility is that anillin KD relieved this negative regulation, leading to the activation of RhoA-dependent myosin enrichment. Alternatively, anillin may directly inhibit the myosin enrichment rate, independently of its positive role in cortical patch formation (inhibitory arrow in Figure 9G). Consistent with both scenarios, normal myosin enrichment rate, as well as ring mechanosensitivity, were restored by the mutation of the RhoA activator *nop-1* (Figures 8 and 9F). Although the biochemical function of NOP-1 remains unclear, NOP-1 localizes to the cleavage furrow and upregulates the cortical localization of RhoA effectors in different axes, such as anillin, myosin, and actin⁵³. This seems unlikely if NOP-1 belongs to one of the downstream axes.

Furthermore, NOP-1 is essential for contractility and RhoA activation in the absence of a centralspindlin component CYK-4^{53,64}. Thus, it is plausible to consider that NOP-1 is upstream of RhoA and positively regulates ring mechanosensitivity (Figure 9G). Finally, we observed that the peak ring eccentricity was increased in ROCK KD (Figure 2D). Therefore, we propose that the RhoA pathway tunes ring mechanosensitivity by modulating the myosin enrichment rate (Figure 9G). Future studies would require structure-function analysis of anillin to uncover the molecular mechanisms underlying the regulation of myosin enrichment rate within the ring.”

Lines 75-76 – “Recent studies show that cortical flow, a concerted flow of cell cortex materials at the cell surface”. Define what cell cortex materials are. This term was taken from Singh et al., but still needs to be explained.

Our original intension was to refer to the materials/components present in the cell cortex. We have modified it to “cell cortex components” if that is the problem. The citation to Singh et al., may seem odd, as cortical flow has been known since 1980s (PMID: 3277283). We don’t think we need to clarify cell cortex as we do not need to clarify cytoplasm or nucleus.

Lines 77-78 “Therefore, we hypothesized that the contractile ring utilizes this mechanism to sense tissue mechanics” – what mechanism? How does cortical flow and compression within a single cell fit a model for sensing tissue? It isn’t clear to the reader how this will influence unilateral ingression.

We have rewritten the entire introduction to address the issues.

Line 86 – confirming vs. confirmed

Now this sentence was deleted due to major rewrite of introduction. Thank you.

Results

It is not clear if all of the knockdowns work, and the extent of knockdown is not shown for any of the conditions. The authors should provide some evidence for this.

We have mentioned a validation method in the revised Method section.

Page 24, Line 539:

“All the feeding vectors were sequence verified and confirmed their effectiveness based either on the reported RNAi phenotypes or the loss of signals of endogenously tagged fluorescent reporters.”

Line 92 – Clarify for the reader that the GFP tag fused to NMY-2 is endogenous (I assume it is the one generated by Dickinson) and if it is homo vs. heterozygous. The authors should also cite studies and/or show that filament assembly is not affected by the tag. Showing actin localization and changes in NMY-2 localization in response to depletion of regulators would support this (e.g., *let-502*, *mel-11*, *ect-2*, *rho-1*).

All the experiments used endogenous myosin II/NMY-2::GFP, which is viable as homozygous. We added the following description:

Page 6, Line 111:

“To understand the mechanism of asymmetric ring closure, we performed high-resolution 4D imaging of endogenously tagged non-muscle myosin II::GFP (NMY-2) in *C. elegans* zygotes (Figures 1B and S1A).”

Page 24, Line 527:

“Other than *itIs37*, all the fluorescent reporters are endogenously tagged and were maintained as homozygotes.”

These CRISPR-edited fluorescent reporters are viable as homozygotes. Since we have used the myosin strain for all the experiments, we believe the suggested experiments are not necessary for interpreting of our data. We believe that the reviewer is referring to myosin minifilament assembly (because myosin II does not affect actin filament assembly but organization), but it is unclear to us how the suggested experiment, “Showing actin localization and changes in NMY-2 localization in response to depletion of regulators (e.g., *let-502*, *mel-11*, *ect-2*, *rho-1*).”, would provide us with valuable insights. If it is of interest, it has already been demonstrated that an increase or decrease of myosin signal occurs after the knockdown of RhoA pathway using this strain (Bell et al., 2020: PMID 32491957).

Lines 95-96 – Add a sentence to explain to the reader what ‘compression-dependent cell rotation is’. I thought that the eggshell still provided some of this external force, or is the eggshell also removed? Also, I didn’t think rotation was much of an issue for the one-cell zygote vs. the two-cell stage.

It is a rotation of the cell induced by artificial external compressive forces. This is observed under normal imaging condition using agarose gel (see Shigh et al., 2019 and Video 1 in Khaliullin et al., 2018). It prevents us from analyzing the leading and lagging edges during division. The eggshell was not removed in our study, except in Figure 7B-H. We have modified the text as follows:

Page 6, Line 113:

“We applied a new imaging method that combines the use of spacer beads and a refractive index-matching sample medium to avoid cellular compression and to improve image resolution⁴⁴, respectively (see Methods for detail). Compression is detrimental as it induces cellular rotation⁴³, hindering our ability to consistently track the leading and lagging edges during division, and it is also known to induce spindle off-centering-dependent asymmetric ring closure³³. These improvements allowed us to capture uncompressed embryonic volumes (30 μm thickness) at 5.6-s intervals with a 1 μm step size, without cell rotation (Figure S1B and Movie S1).”

Line 106 – ‘degrees of leading-edge and lagging-edge advancement towards the center’. More explanation is needed to explain how this measurement was determined – why over 2RO?

Please see the left schematic in Figure 1C. We have changed “advancement” to “ingression” for improved clarity.

Lines 122-123 – “ Δt , was determined by the delay between the leading and lagging edges reaching 10% advancement relative to the initial ring diameter”. This is not clear, perhaps just say onset of ingression?

Please see Fig. 1F. We have modified the text as follows.

Page 7, Line 143:

“The time lag, Δt , was determined by the delay of lagging edges reaching 10% ingression relative to the initial ring diameter (Figure 1F).”

Line 124 – typo ‘R closure’

This is corrected in the revised manuscript.

Lines 128 – 133 – It isn’t clear why the authors divided the time lag by velocity. Their argument is that a ring that closes more slowly would have a longer time lag, but isn’t that the point, since velocity is determined by the change in distance over time? To divide the delta time by velocity (which is also determined by time) seems to put more weight on the time component. This is followed by the conclusion that both velocity and time lag control asymmetric ring closure, which seems confusing. I suspect I am missing something in their explanation.

We multiplied them, as stated in our manuscript, not divided.

Lines 134 – 141 – The rationale for knocking down the genes they chose is not clear and seems to be strictly based on Singh et al. Some of the genes are crosslinkers while others regulate contractility, which could have different effects on cortical properties and mechanics. For example, what is the function of POD-1/coronin and why did the authors knock this gene down? The authors could consider grouping genes according to their function e.g., actin crosslinkers: ani-1, septin, alpha-actinin (and pod-1?) vs. actomyosin contractility: let-502, ect-2, rho-1, nop-1. Also, some of these genes can also affect anterior-posterior polarity and cortical flows that are not ring specific. How can the authors separate these from ring-specific phenotypes?

These genes are well-known actomyosin regulators, known long before the 2019 paper. Data driven hypothesis formulation does not need full coverage of actomyosin regulators, and it influenced our selection of genes (e.g., as long as we have several conditions with asymmetric and symmetric closure, we can perform this analysis). In our original manuscript, we omitted conditions resembling wild-type (partly due to the space limit), but we have now included them. Additionally, we have provided explanations for each component, as mentioned in our previous comment, and incorporated several actomyosin regulators suggested by the reviewers.

Lines 160 – 161 – Clarify what is meant by “we hypothesized that the delay is due to the feedback loop having lower activity at the lagging edge than at the leading edge”. Lower activity of what? Is this in reference to ring-directed flows? Do they mean that the feedback starts lower and stays lower in one region? Why would it be lower to begin with? The authors should distinguish between what could initiate vs. propagate asymmetry – e.g., if there is asymmetry to begin with, then it will be maintained by feedback.

We clarified this point in our revised manuscript as follows and added Figure 4G:

Page 13, Line 278:

“As ring-directed flow is driven by ring closure, one plausible mechanism that inhibits ring-directed flow at the lagging cell cortex is the tug-of-war between the contractile cell cortex and the constricting ring. Therefore, we hypothesized that circumferential compression inhibits ring-directed flow, thereby suppressing a flow-dependent positive feedback loop in ring assembly (Figure 4G).”

Figure 3A legend – ‘Resulted myosin enrichment to the ring further facilitates ring constriction’ change

to 'Myosin enrichment in the ring facilitates constriction'. Also, more information should be provided to the reader to understand what is shown in the figure.

We have added Figure 1A to aid in understanding the schematic in Figure 3B. Additionally, we have revised the figure legend to align with the previously established knowledge (and deleted speculations) as follows:

Figure 4 legend:

"Cortical flow-dependent positive feedback model in contractile ring assembly. Ring-directed cortical flow delivers cortical materials to the equator, enriching myosin and promoting ring constriction. Ring constriction likely triggers ring-directed flow, as suggested by previous in silico analyses."

Figure 3. Why do the authors use normalized time on the X-axis vs. real time? Also, I assume the first dotted line is leading furrow onset?

We used normalized time in some parts because it is useful when comparing the two conditions with significantly different ring closure velocities. Times are relative to cytokinesis onset.

I am struggling to understand the role of the circumferential vectors the authors call 'convergence' when they are imaging the lagging edge quite late in cytokinesis (e.g., the ring is >50% closed), so it is not surprising that the vectors will shift down vs. across.

We have added Figure 3 to offer additional 3D information regarding these cortical flows.

For Figure 4, the different conditions need to be explained to the reader.

We have reorganized the figures for clarity, moving the previous Figure 4 to Figure 8. Additionally, we have improved the labeling of data based on the reviewers' suggestions.

Also, "We found that conditions with asymmetric ring closure (control, ani-1;nop-1, and let-502) and symmetric ring closure (ani-1 and nop-1) exhibited a reduction and no change in ring directed compression, respectively (Figure 4C)." Does this mean that nop-1 closes symmetrically? This contradicts their measurements of eccentricity for the nop-1 mutant in Fig. 2D.

This was a typo and nop-1 should be removed from the text. We have corrected this error in the revised manuscript.

Lines 211-212 – "Since control and ani-1(RNAi) displayed a similar ring closure velocity (Figure 2F), the observed cortical relaxation cannot be attributed to the ring closure rate." This is not clear, explain what is meant by cortical relaxation.

We have removed several unnecessary speculations, including this sentence, to avoid confusion. Now this data is presented in Figure 8 instead of Figure 3, which is a better context without the need of speculations.

Lines 221-222 "We hypothesized that any non-ring-directed cortical compression, like the circumferential-axis compression we observed, inhibits ring-directed cortical flow." I am confused. Up until this point, the earlier text made it sound like the circumferential axis compression in Figure 3 was caused by constriction (which is ring-directed). I understand that they also see that contractility outside

the ring can have an impact on ring-directed flows, but this sounds confusing. Importantly, the figure shows an ROI in the equatorial plane.

To better illustrate circumferential compression, we have added Figure 3. We believe that this addition, along with various text changes (too many changes to highlight here), has addressed the concerns.

Figure 5 – The data needs to be more clearly explained. I understand what the authors are trying to show, but the way it is presented it is challenging to follow and it is not clear to the reader what is being shown and why. The images are not clearly explained in the legend or in the main text. The imaging locations, timing, etc is not described. For example, in Figure 5A, where is the leading vs. lagging cortex for the *act-5* mutant, and if symmetric, then show the reader where the ring is being measured. Figure 5B, what is being shown here? control zygote – stabilization of what? Figures 5C and D, where is this taken from in the embryo? What does ‘area behind the ring’ mean? Italicize gene names.

Now we have added further explanation to the figure and legend as follows.

Figure 5 legend:

“Figure 5. Cortical compression suppresses ring-directed cortical flow.

(A) Kymographs of cortical myosin. Yellow boxes in the top images were used to generate kymographs. Periods of ring-directed flow and no flow were indicated by colored boxes next to the kymographs. *act-2(gof)* exhibits high cortical contractility and increased no-flow periods. (B) Ring-directed cortical flow in control zygotes. The leading edge exhibits consistent flow, while lagging edge flow is interrupted by a no-flow period. (C) Ring-directed cortical flow in *act-2(gof)* mutants. Movement of a myosin focus during contraction and relaxation of the region immediately anterior to the ring is shown. (D) Averaged images of myosin foci during cortical contraction and relaxation in *act-2(gof)* mutants. (E) Mechanism of compression-induced inhibition of ring-directed cortical flow. Ring-directed flow is impeded by a tug-of-war between the constricting ring and the contractile cortex. Times are relative to cytokinesis onset. Scale bars, 10 μm .”

Also, they should refer to the measurements shown for the *act-5* mutant in Figure 2. It seems to lose eccentricity, so does this support their model or not? How do they interpret this finding with respect to external compression and potential impact on the ring-directed flow?

We have added explanation in the text as follows:

Page 14, Line 306:

“However, the contractile rings of the *act-2* mutant frequently displayed oscillatory movement during constriction and exhibited a lower mean peak eccentricity, likely due to wandering (Figures 6A and 2D).”

Also in reference to Figures 5 and 6, in the main text, the authors could more clearly state that non-ring directed compression (e.g., ectopic contraction) can inhibit ring-directed flow. This concept is logical to follow – competition from different contractile events.

Thank you for the comment. We have revised Figure 5-6 and modified the text to improve clarity.

Figures 5 and 6 could be combined, with some of Figure 5 being moved to supplemental data.

We attempted the suggested change during the preparation of the original manuscript, but it was not effective in our hands.

Lines 236-237 – what is ‘behind the ring’?

This terminology was used in other papers, but we changed it to “adjacent to the ring” for better clarity.

Line 243 – delete ‘than’ (‘than the control)

Corrected in the revised manuscript.

The authors could gain more support for their model by co-depleting *ani-1* in the *act-5* mutant, where the ‘anchoring’ of actomyosin filaments mediated via *ani-1* could be disrupted. Or do they mean something else?

Here, we originally meant the anchoring of the ring to the cell cortex. We deleted the term “anchor” from this part of the text to avoid confusion.

Figure 6 – The trajectory of the centroids is hard to see as they are quite small, could the authors make these larger? Also, why measuring acceleration vs. displacement for the edge and center? No explanation is provided to the reader on the rationale for some of the methods used for measuring parameters of the ring.

Please see the ring trajectory shown by the perimeter instead. We attempted using thicker lines, but it worsened the appearance. The ring perimeters are shown in blue and yellow when the ring is moving towards the bottom and top, respectively.

Figure 7 – There is discrepancy between the text vs. what is shown in the figure. For example, two beads are shown in the figure, and this seems to cause the greatest impact on eccentricity, but in the text, there is no mention of this. The authors should speculate (in the Discussion) on why they needed two beads where each one is larger than an individual P1 cell to cause eccentricity that still isn’t fully restored to control levels.

This experiment demonstrated that the inhibition of ring-directed flow induces asymmetric ring closure in symmetrically dividing cells. Our goal here is not to fully restore the wild-type level of eccentricity, which is influenced by the physical adhesion strength of the bead. Although the size of the bead is larger than that of the P1 cell, the contact size of two-beads is comparable to that of intact embryos, as shown in Figure 7A and 7F.

We have modified the text as follows:

Page 16, Line 344:

“A single bead attachment resulted in a slight increase in the average peak eccentricity, although this change was not statistically significant (Figure 7C, $p = 0.11$). This result may be attributed to the smaller contact area compared to *in vivo* conditions (Figures 7A and 7D) and the normal ring-directed flow at the cell cortex distal to the bead (Figure 7E). Notably, the attachment of the two 30 μm beads increased peak eccentricity, and about 50% of the cells exhibited a level of ring eccentricity comparable to that of the intact AB cell *in vivo* (Figure 7C, F–H). Thus, we conclude that mechanical obstruction of ring-directed cortical flow is sufficient to induce asymmetric ring closure.”

Also, for Figure 7 – What do D and P refer to? Are two beads ns vs. an intact cell or still different? Stats not shown for this comparison.

D and P indicate distal and proximal cell cortex, as shown in Figure 7D. We have provided an explanation in text and figure legend. A comparison between two beads and intact embryos does not give us with valuable biological insights. For the record, the two-beads samples have reduced eccentricity compared to intact embryos. There are numerous differences between bead-attached embryos and intact embryos, such as variations in physical adhesion strength, cell shape, attachment firmness, potential technical errors, and differences in contract area. Thus, comparing intact embryos with bead attached embryos is inappropriate. In this case, meaningful comparisons include intact vs isolated, isolated vs one bead, isolated vs two beads, and one bead vs two bead.

Page 16, Line 340:

“As shown in our previous study, attachment of the 30 μ m diameter beads reduced ring-directed flow in the area proximal to the bead attachment site (Figure 7E) ⁵⁴. The response scaled with the bead diameter, indicating that passive mechanical resistance imposed by adhesive beads limits ring-directed flow (Figure 7E). A single bead attachment resulted in a slight increase in the average peak eccentricity, although this change was not statistically significant (Figure 7C, $p = 0.11$). This result may be attributed to the smaller contact area compared to *in vivo* conditions (Figures 7A and 7D) and the normal ring-directed flow at the cell cortex distal to the bead (Figure 7E). “

Figure 8 – typo – called Figure 7 on the figure.

We have corrected this in the revised figure. Thank you.

Lines 292-295 – clarify what the reader is meant to infer from the pod-1 RNAi embryos which lose asymmetry in P0 but not in AB. What about *mlc-4*? The authors indicate that the loss of circumferential convergence could explain this, but not why this doesn't occur in AB. Further, they write ‘These results suggest that the contractile ring in pod-1(RNAi) retains its mechanosensitivity but lacks the P0 mechanical cue.’ I am not sure what they mean by this statement.

In the revised manuscript, we have emphasized the role of anillin. We have provided an explanation of the interpretation of the pod-1/coronin phenotype in zygotes within this context. *mlc-4*/MRLC RNAi results in a failure of cytokinesis at the two-cell stage, preventing us from generating this data set. Figure S5B in the revised manuscript summarizes the eccentricity at the two-cell stage.:

Page 16, Line 354:

“Our analysis sheds light on the mechanism of contractile ring mechanosensation, which requires the mechanical inhibition of the ring-dependent generation of ring-directed flow (Figure 9H; cortical flow pathway and mechanical cue). Consistently, the lack of circumferential compression, as measured by the deformation rate of the lagging cortex, resulted in symmetric ring closure in zygotes of *ect-2*, MRLC, and Coronin knockdown (Figures 8C and 2D).”

Page 17, Line 376:

“Consistently, anillin KD also reduced ring eccentricity during the presumable adhesion-dependent asymmetric ring closure of the two-cell stage AB cell, whereas Coronin KD underwent normal

asymmetric ring closure (Figure S5B). Thus, anillin is specifically required for contractile ring mechanosensation.”

One plausible hypothesis for the lack of asymmetric closure in the isolated AB cell is the slower speed of contractile ring closure. A previous study shows that the larger cells tend to have faster ring closure rates (PMID:19490897). Given that ring closure velocity is also an important factor contributing to asymmetric ring closure (Figure 2), it is possible that a slower ring closure rate in the AB cell cannot generate sufficient circumferential flow, similar to what is observed in *ect-2*, *coronin*, and MRLC knockdown in zygotes.

When referring the loss of mechanosensitivity for *ani-1* RNAi – do they mean to say that this is for both P0 and AB cells? Clarify for the reader.

Please see above response.

Figure 8 – The authors should refer back to and/or show velocity measurements vs. correlative plots (only some are shown earlier in Figure 2). The idea that myosin is enriched in *ani-1* RNAi embryos, yet closure occurs at the same rate/velocity demonstrates that the myosin enrichment alone may not be sufficient, which has been shown by other studies – breadth or distribution could be a more meaningful parameter, but it wasn't measured here. The authors should comment if they also measured breadth or if their calculation considers this parameter as well.

As shown in Figure 9A-C, there is a consistent correlation between cortical flow, myosin enrichment rate, and ring closure velocity except for anillin KD. This aligns with the concept of a cortical flow-dependent feedback loop in ring assembly. However, this is distinct from a previous observation mentioned by the reviewer that “myosin enrichment alone may not be sufficient, which has been shown by other studies”. As seen in Figure 9B, myosin enrichment rate effectively explains ring closure velocity except in the case of anillin KD.

Please note that the myosin enrichment rate we quantified differs from myosin intensity, whereas the reviewer's comment is on myosin intensity. We measured the total myosin signal within the ring, as shown in Fig. 9D and 9E, and calculated the enrichment rate by determining the slope between local minima immediately after cytokinesis onset and the maximum myosin signal peak. Notably, the total ring myosin signal does not correlate well with ring closure velocity (Figure 9A and 9E) (e.g., *coronin* RNAi results in a total myosin signal similar to the wild-type level but the velocity is very slow), which aligns with previous observations.

Discussion

In general, there are a few pieces of data that the authors should speculate on in their discussion, and include some of the caveats of their study.

We have rewritten the discussion and included the limitations of the study as follows.

Page 21, Line 467:

“Future studies would require structure-function analysis of anillin to uncover the molecular mechanisms underlying the regulation of myosin enrichment rate within the ring.”

Page 21, Line 470:

“Our study leveraged a simple model system without neighboring cells and extrinsic forces.”

Page 22, Line 475:

“Hence, our model should be valuable in understanding the mechanosensation of the contractile ring in multicellular contexts. To achieve this goal, it is imperative to integrate other mechanosensitive mechanisms into this model, such as the mechanosensitive regulation of myosin in neighboring cells during division^{66,67}. “

Also, there is also an issue in how their model is presented (Rho-dependent ring vs. flow-directed ring), since cortical flows are still driven by RhoA and these are not necessarily separable. I understand what the authors are trying to say, but they should consider different terminology and/or be very clear in their explanation.

We have added Figure 1A to provide a clear depiction of the relationship between RhoA and cortical flow pathways at the beginning of the manuscript. While cortical flow is influenced by RhoA, it is more directly associated with the contractility gradient (PMID: 20852613). Thus, we have added an arrow extending from myosin II and actin toward the contractility gradient.

To enhance clarity, we have separated the model of mechanosensation and the model of mechanosensitivity tuning clearly within the discussion. The mechanosensation model, which is mediated by the cortical flow pathway, does not require an explanation of the RhoA pathway.

The Figure in 8G needs some modification. They show a negative arrow from ANI-1 to RhoA. This is not supported by their data, and if anything has been shown to have positive vs. negative feedback to RhoA. I realize that this is based on the Bell et al. paper where ANI-1 can feedback to RGA-3/4, but this could facilitate RhoA turnover to make more RhoA-GTP. The only evidence they have for this arrow in this study is based on the higher/more symmetric levels of myosin in the ring, but this doesn't mean there is more active myosin in the system (see earlier comments). One way to deal with this contradiction is to just remove the blocked arrow from the figure. They can still have ANI-1 playing a central role in their mechanosensing model in H, where it makes more sense in having more localized effects on myosin vs. global.

In the original manuscript, we constructed a model based on the evidence presented in the Bell et al., paper, because the negative feedback might offer a better explanation for the rescue of the ani-1 phenotype by the nop-1 mutation. We want to clarify that the reviewer's statement “ANI-1... make more RhoA-GTP” may have been a misunderstanding of their paper. According to the paper, ANI-1 inhibits Rho-1 through GCK-1/CCM-3, as GCK-1/CCM-3 “promotes” the recruitment of RGA-3/4 to the cell cortex. To reiterate, RhoGAP/RGA-3/4 serves as negative regulators of RhoA, and ANI-1 enhances the function of RGA-3/4, thereby inhibiting RhoA. Therefore, we do not consider our original interpretation was contradictory to this evidence.

In the revised manuscript, we have introduced an alternative model in consistent with the reviewer's suggestion, as follows.

Page 20, Line 446:

“Our analysis suggests that components of the RhoA pathway tune contractile ring mechanosensitivity (Figure 9G; left box). We demonstrate impaired mechanosensitivity in anillin KD embryos, in which myosin was too rapidly enriched in the contractile ring (Figure 9). This observation is somewhat counterintuitive, considering the role of anillin as a scaffolding protein for actomyosin components and its involvement in the formation of large cortical patches, containing myosin⁷². Recent research has

shown that Anillin-binding proteins GCK-1 and CCM-3 negatively regulate RhoA activity by promoting RhoGAP cortical localization⁷³. Therefore, one possibility is that anillin KD relieved this negative regulation, leading to the activation of RhoA-dependent myosin enrichment. Alternatively, anillin may directly inhibit the myosin enrichment rate, independently of its positive role in cortical patch formation (inhibitory arrow in Figure 9G). Consistent with both scenarios, normal myosin enrichment rate, as well as ring mechanosensitivity, were restored by the mutation of the RhoA activator *nop-1* (Figures 8 and 9F).”

Lines 373-374 - “this variability suggests that the mechanosensitivity of contractile ring can be tuned differently during development, enabling tissue-specific morphogenic cytokinesis.” What is morphogenic cytokinesis?

We defined this term in the revised version of introduction.

Page 3, Line 39:

“The cytokinetic contractile ring physically partitions the dividing cell during cell division, but it also plays pivotal roles in morphogenesis by regulating its position and function along the body axis¹⁻³. This is evident in processes such as asymmetric cell division and epithelial morphogenesis, where the position and closure of the ring are asymmetrically regulated to control the size, shape, and arrangements of daughter cells. Although previous studies have suggested that both intracellular and extracellular mechanics influence this “morphogenetic cytokinesis”⁴⁻⁶, the mechanism by which the contractile ring senses mechanical cues and modulates its function in response remain largely unexplored.”

Reviewer #2 (Remarks to the Author):

The manuscript by Hsu et al. explores the mechanism of mechanosensation in the contractile ring. As a model system, the authors use the asymmetrically closing ring of the early *C. elegans* zygote. Asymmetric ring closure has been observed in a number of metazoan cells but the underlying mechanisms are incompletely understood. Previous studies implicated Anillin and other cortical proteins in asymmetric ring closure in *C. elegans*. A positive feedback model during ring constriction that relies on cortical flows and Myosin enrichment in the contractile ring has also been proposed previously. Here, Hsu et al., extend this model by carefully measuring ring closure dynamics in wild type and mutant worms. The authors developed a new live cell imaging method to measure ring dynamics, furrow ingression and cortical flows in both wild type and mutant situations. The authors propose that asymmetric ring closure is regulated by a mechanical feedback loop whereby local mechanical cues suppress ring-directed cortical Myosin flows which reduce compression-dependent Myosin enrichment in the contractile ring.

The manuscript convinces with detailed measurements and elegant genetic manipulations. The authors also use beads to manipulate non-ring directed cortical flows. The underlying questions and findings are of considerable relevance and this study could make a significant contribution to the field. However, in its current form, the manuscript is difficult to read mostly because the authors use several mechanical concepts that are either poorly defined/explained or used interchangeably. For instance, the connection between cortical flows, convergence and compression is not intuitively clear and the manuscript uses some terms very suddenly without much introduction or explanation. Maybe a

supplemental figure that summarizes these concepts in graphical form would help interpret the data better. Also, I find it difficult to see how these concepts are related to asymmetric ring positioning because they appear to apply more generally to furrow formation and ingression dynamics. Addressing the issues below might help with this general comment.

We appreciate the reviewer's insights into our manuscript and pleased to see positive comments. Now we have rewritten manuscripts, reordered data, and acquired additional data to provide a more comprehensive explanation of our results.

Critiques:

(1) Line 75-78: Recent studies show that cortical flow, a concerted flow of cell cortex materials at the cell surface, facilitates contractile ring assembly through cortical compression^{37–39}. Therefore, we hypothesized that the contractile ring utilizes this mechanism to sense tissue mechanics.

The concept of "cortical compression" should be briefly introduced here. Also, the subsequent hypothesis needs to be explained in more detail. What type of tissue mechanics do the authors envision? This is particularly relevant as the model system, the P0 zygote is not surrounded by any neighboring cells.

We have rewritten this part and explained cortical compression more clearly.

Page 4, Line 74:

"*C. elegans* zygotes offer a unique model system to study unilateral cytokinesis because they lack both adherens junctions and a transversely off-centered spindle⁹. In this system, contractile ring components concentrate at the leading edge, forming a structurally asymmetric ring⁹. The formation of this structurally asymmetric ring is regulated by contractile ring components anillin and septin via unknown mechanisms⁹. Recent studies using these zygotes have shown that cortical flow, a concerted movement of cell cortex components at the cell surface, facilitates contractile ring assembly through a mechanical process^{39–41} (Figure 1A). Cortical flow is driven by a gradient in cell cortical contractility⁴², and during cytokinesis, it is oriented from the polar region toward the ring (hereafter called ring-directed flow). These studies proposed that ring-directed cortical flow from opposite poles compresses the gel-like cortex, thereby enhancing and aligning myosin concentration and actin orientation, respectively^{39–41}. Interestingly, the amplitude of ring-directed cortical flow differs between the leading and lagging cell cortex³⁹, and artificial cellular compression via external forces influences both the pattern of cortical flow and the degree of ring closure asymmetry^{43,44}. Nevertheless, it remains unknown whether asymmetric closure requires 1) ring-directed flow, and 2) mechanical regulation, given the involvement of multiple confounding factors, compression-dependent cellular rotation occurring in a commonly-used imaging condition which prevents us from analyzing leading and lagging edges as they rotate⁴⁵, the lack of comparative 4D analysis in mutants of actomyosin regulators, and the challenges in simultaneously imaging the cell cortex and the ring.

We hypothesized that asymmetric ring closure in *C. elegans* involves asymmetric regulation of the cortical flow through mechanical processes..."

(2) How is the parameter space in Figure 2F defined?

The original text aimed to depict the binary separation of concentric and eccentric phenotypes in this xy coordinate system. We have now added clarification, describing that there is a tendency that for

cells to undergo asymmetric ring closure with higher the time lag and increased ring closure velocity, as follows:

Page 9, Line 184:

“Furthermore, when examining the relationship between ring closure velocity and time lag, we found that conditions with asymmetric and symmetric closures are separated within the parameter space (Figure 2F). Cells exhibiting slower ring closure, as well as shorter time lag, tended to display defective unilateral cytokinesis.”

(3) The cartoon in Figure 3A is helpful but it would be more informative if it reflected the actual asymmetrically ingressing furrow situation.

The original cartoon summarizes the known compression-dependent feedback pathway of contractile ring assembly. In the revised manuscript, we have added Figure 4G to depict the asymmetric activities of this pathway.

(4) Line 168: Given that the contractility gradient continues to exist during ring assembly and closure, the ring should consistently pull the cell cortex from the polar region.

What is the evidence for this statement?

Mayer et al., reported that the contractility gradient drives cortical flow during the cell polarization process (PMID: 20852613). In the context of cytokinesis, equatorial region is known to be more contractile compared to the polar region. Consistently, a computer simulation shows ring-directed flow in response to higher contractility at the equator compared to the pole. However, whether the ingressing edge of the contractile ring pulls cell cortex to generate ring-directed flow remained unclear before our study (this was not clearly stated in the original manuscript).

To understand the relationship between the ingressing ring edge and the cortical flow, we compared the speed of ring edge ingression with ring-directed cortical and found a good correlation between them (Figure 4C). In the revised manuscript, we further used a *zen-4*/MKLP1 knockdown, in which the contractile ring partially constricts but eventually regresses, resulting in cytokinesis failure. Strikingly, during ring regression, we observed pole-ward movement of the cortical flow (Figure 4D). We found a strong correlation between the timing of ring ingression/regression transition and the timing of the transition from ring-directed to pole-ward flow. These results suggest that the ring edge pulls the nearby cell cortex, inducing ring-directed cortical flow.

Page 11, Line 236:

“Next, we examined the mechanisms responsible for generating ring-directed flow. Cortical flow is driven by a gradient in actomyosin contractility⁴², and a computer simulation have indicated that higher contractility at the equator, compared to the pole, drives ring-directed flow⁶¹. Indeed, inhibition of cortical contractility at the polar cell cortex facilitates ring-directed flow^{25,27}. Nevertheless, the relationship between ring-directed flow and ring closure has remained unclear due to the absence of 4D analysis. We plotted velocities of ring edge ingression (Figure 4C; solid lines) and ring-directed cortical flow (Figure 4C; dotted lines) and found that they correlate well both at the leading and lagging edges (leading edge: Pearson’s $r = 0.77$, $p < 0.0001$, lagging edge: Pearson’s $r = 0.89$, $p < 0.0001$). We also compared the velocity of ring edge ingression and ring-directed flow in *zen-4*/MKLP knockdown. MKLP is a component of centralspindlin, and its knockdown leads to ring regression midway through cytokinesis. We observed ring-directed flow during ring constriction; however, when the ring started to regress, cortical flow moved away from the ring (Figure 4D). The timing of transition

between ingression and regression of the ring strongly correlated with transitions between ring-directed and pole-ward flow (Pearson's $r = 0.98$, $p < 0.0001$; Figure 4E). Therefore, we conclude that the invaginating ring edge consistently pulls the nearby cell cortex, generating ring-directed cortical flow. These findings shed light on the origin and function of ring-directed flow and provide further support for the flow-dependent positive-feedback model in ring assembly."

(5) The connection between cortical flow and axis convergence is not well explained in the manuscript. It makes it difficult to conclusively interpret the results shown in Figure 3 and 4. For instance, the statement from line 189 "These results suggest that circumferential-axis cortical flow negatively correlates with ring-directed cortical flow" is not really supported by the data.

In the revised manuscript, we have substantially expanded our explanations, incorporating new data. In response to the reviewer's suggestion, we have removed "negative correlation" from the text to eliminate any confusion.

Page 10, Line 202: Explanation of cortical flow measurement

"We next analyzed cortical flow as it might relate to a relative furrowing delay at the lagging edge. As reported previously³⁹, we observed ring-directed cortical flow at both the leading and lagging edges, with a delay at the lagging side (Figure 3A and Movie S2). To quantitatively measure cortical flow dynamics, we employed particle image velocimetry (PIV)⁵⁸ (Figures 4A and S4, and Movie S3), and confirmed that ring-directed flow is indeed delayed at the lagging cell cortex (Figure 4C; dotted lines). During this delay, we identified a previously uncharacterized, orthogonally oriented flow, which we termed circumferential cortical flow (Figure 3A, bottom panel and Movie S2)."

Page 12, Line 265:

"To better measure the flow dynamics at the equatorial region, we calculated the convergence of flow vectors along different axes (Figure 4A). Here, when the overall influx of flow in the equatorial region is greater than the efflux, we will obtain a positive convergence value. At the leading cell cortex, the convergence of ring-directed flow increased rapidly and reached its peak before the onset of lagging edge furrowing (Figure 4F; left). Conversely, the convergence of ring-directed flow in the lagging cell cortex decreased shortly after cytokinesis onset and reached a local minimum before the onset of lagging edge furrowing (Figure 4F; left, Figure 3E; $t_n = 58$). These results indicate that ring-directed flow is inhibited at the lagging cell cortex. During this period, we observed an increase in circumferential axis convergence at the lagging cell cortex (Figure 4F; right). Taken together, these results indicate that ring closure induces circumferential cortical flow (note: it's shown in Figure 3) at the lagging edge, which coincides with the inhibition of ring-directed flow."

(6) Line 197: "After the onset of furrow ingression, convergence at the surface represents the sum of cortical compression and ingression."

What data or previous findings support this statement?

In fluid mechanics of incompressible fluid, such as water, a negative divergence (= convergence) signifies the compression or movement of fluid toward a point, often referred to as a sink. During cytokinesis, the cell cortex exhibits movement toward the deeper cell center, akin to the movement of sea water toward the ocean floor. Before cortical ingression, it can be assumed that negative divergence results from cortical compression. This approach is reminiscent of the method used by Armon et al., to estimate cellular contraction and relaxation (PNAS 2018, PMID: 30309963). Additionally, we have introduced Figure 8C, where we directly quantified the strain rate of the

equatorial cell cortex by measuring the deformation of the cortical area, using myosin foci as a positional markers.

Page 17, Line 364:

“To characterize spatio-temporal changes in cortical mechanics, we measured cortical convergence of flow vectors. Convergence and divergence of flow at the surface inform cortical compression and relaxation, respectively, provided there is no involution or extrusion^{65,66}.”

(7) Line 229: The concept of “relaxation-induced” ring-directed cortical flow needs to be introduced.

We have added new Figure 3 to provide a clear illustration of ring-directed flow. Additionally, in Figure 5, we have presented new schematics (Figure 5E) to demonstrate the relationship between cortical compression/relaxation status and ring-directed flow. We have also modified the text as follows:

Page 13, Line 289:

“By tracking myosin foci during cortical contraction and relaxation, we found that ring-directed flow was suppressed during contraction and **induced during relaxation** (Figures 5C-D). Ring closure is necessary for **the relaxation-induced ring-directed cortical flow**, as similar flow was not observed post-cytokinesis (Figure 5D; right). These results suggest that non-ring-directed cortical compression inhibits ring-directed cortical flow at the lagging cell cortex (Figure 5E).”

(8) It is not clear why the bead experiment was performed in the AB cell. The mechanisms for asymmetric furrowing could be different in this cell type. To make the manuscript flow more coherently, it would be beneficial to provide bead data from the P0 zygote to correlate these experimental data with the other findings of the manuscript.

Now we have reorganized the figures, incorporating additional information to clarify our logic. The data presented in Figure 5 and Figure 6 suggest that cortical compression inhibits ring-directed flow and ring edge ingression. However, it is possible that these processes are redundantly regulated, as shown in Figure 6I. To confirm that the inhibition of ring-directed flow is sufficient to induce asymmetric ring closure, we used isolated cells that normally undergo symmetric ring closure. Therefore, the use of symmetrically dividing ring was essential for this purpose.

Additionally, isolating the P0 zygote is technically challenging. Since our data suggest that circumferential axis compression is generated by ring closure (Figure 3), it is very likely that isolated P0 zygotes undergo asymmetric closure. The AB cell, an immediate daughter of the P0 zygote, and its sibling P1 both undergo symmetric and asymmetric ring closure, without and with bead attachment, respectively (Supplementary review figure 1). These data suggest that cell fate is not an important contributing factor.

Supplementary review figure1. P1 cell undergoes unilateral cytokinesis in the presence of beads

Left: Isolated P1 cell undergoes symmetric closure. Right. The attachment of beads induces asymmetric closure.

(9) Line 344: In our model, intracellular and intercellular mechanical cues locally limit ring-directed cell cortex pulling at the lagging cell cortex. This statement is unclear as the main model used here, the P0 zygote, has no cell neighbors. Thus, it is not clear what is meant by intercellular mechanical cues.

We have removed unnecessary speculations from the manuscript, including the one mentioned in this comment. In the original manuscript, we found that adhesive beads limit ring-directed flow, and the term “intercellular mechanical cues” meant adhesion *in vivo*.

(10) The manuscript (and figures) should more carefully distinguish between the leading and lagging edge. For instance, in Figure 4, it is unclear whether compression data are shown for the leading or lagging furrow.

In the revised figures, including Figure 8 (old Figure 4), we have provided clearer labels for the leading and lagging edges.

(11) How do the authors define “cytokinesis onset”? This is relevant to correctly understand the timing of events.

The information was included in the main text of the original manuscript as shown below.

“Due to the natural variations in cytokinesis timing, we aligned the time series data relative to the time around 10% contractile ring closure (Figure S1D). We also defined cytokinesis onset based on the extrapolation of an initial linear part of the ring closure curve to 0 (Figure S1D).”

As pointed by the reviewer 3, the linear part was subjectively determined. In the revised manuscript, we show that this estimation method is robust and not influenced by the subjective selection of the linear part as shown in Figure S2.

(12) What is the mechanism for the circumferential-axis directed flow?

Now we have added Figure 3 to visualize cortical flow dynamics in 3D space. Notably, circumferential axis flow occurs as the contractile ring closes and compresses the lagging cell cortex, as depicted in Figure 3B.

(13) I would encourage the authors to provide a graphical model (similar to Figures 3A, 4F, 5E) at the end of the manuscript. That would also allow to include the effect of the different mutants more clearly.

In the revised manuscript, we have added an extensive number of graphical summaries of our data (Fig1A, Fig 2B, Fig4C, Fig. 4G, Fig. 5E, Fig. 6G-I, Fig. 8C-F, Fig. 9A-C, G, H). Additionally, we have added a new Figure 3 to provide a clearer visualization of cortical flow in 3D space.

(14) The y-axis in Figure 5C contains a typo.

We have corrected the typo in the revised manuscript. Thank you.

Reviewer #3 (Remarks to the Author):

Rou Hsu and coworkers investigate the dynamics of cytokinesis in the early worm embryo. These dynamics have been studied intensively but this latest work uses higher temporal resolution and an improved imaging setup. Furthermore, the analysis focuses on how “mechanical cues” influence the contractile ring. The work itself along with the presentation are very well done with clear main figures and supplemental panels that nicely outline the workflows used. The first part of the manuscript makes some interesting and important observations, that cortical flow asymmetry accompanies ring eccentricity and that transverse flow occurs on the lagging cortex. They argue that these transverse flows lead to “cortical compression” that is responsible for the delay that leads to ring eccentricity but that this occurs without inhibiting ring constriction, possibly because of compensation. As part of a test of this model, they use beads to perturb cortical flows and induce eccentricity in a system that normally lacks it. Interestingly, they argue that Anillin promotes ring eccentricity by repressing cortical myosin activity. Overall this is a well executed and important study that makes important advances in our understanding of the relationships of cortical flows, cortical mechanics, and cytokinetic dynamics. I recommend publication with the following additional considerations.

We appreciate the reviewer’s insights and are pleased to see the positive feedback on our manuscript. We have now rewritten the manuscript to offer a more comprehensive explanation of our results.

* The “linear part” in figure S1D concerns me. Since the data are highly nonlinear it seems like it would be challenging to find the appropriate section of data to apply the linear fit. I’m unsure how much this decision would affect the analyses derived from these data but it certainly seems like it could be a very significant effect.

To assess the robustness of our estimation method, we conducted tests using different numbers of data points, as shown in new Figure S2. In addition to simple linear regression, we also used second-order polynomial fitting. In both cases, variations in the number of data points did not affect the estimation of cytokinesis onset, with linear regression performing better overall. We also made attempts to fit the entire time series data using sigmoidal curve fitting, but this approach failed in multiple instances. We believe that the impact of data point selection on the estimation becomes more significant when the temporal resolution is lower than our analysis.

* Line 160 - it would be helpful if the authors explained what they mean by “the feedback loop having lower activity” in more detail.

We clarified this point in our revised manuscript as follows and added Figure 4G:

Page 13, Line 280:

“As ring-directed flow is driven by ring closure, one plausible mechanism that inhibits ring-directed flow at the lagging cell cortex is the tug-of-war between the contractile cell cortex and the constricting ring. Therefore, we hypothesized that circumferential compression inhibits ring-directed flow, thereby suppressing a flow-dependent positive feedback loop in ring assembly (Figure 4G).”

* Some of the graphs apparently plot normalized values (which would lack units) but are labeled as “AU”. It would be more helpful to provide some information about the normalization.

We have made corrections to the units of measurement in several graphs to accurately reflect the data, and we have introduced a new section in the Materials and Methods to provide detailed explanation of these units.

Page 13, Line 708:

“Units of measurement

In several figures, we presented data in arbitrary units. The degree of ring closure was shown, with a value of 1.0 indicating complete ring closure. On the other hand, eccentricity reaches 1.0 when the distance between the centroids of the initial and current ring equals the radius of the initial ring (the theoretical maximum possible ring off-centering at the end of cytokinesis). Cortical flow and flow convergence data were displayed in $\mu\text{m}/\text{sec}$ and sec^{-1} , respectively.”

* Figure 8 is mislabeled as Figure 7

The numbering was corrected in the revised manuscript. Thank you.

REVIEWERS' COMMENTS

Reviewer #1 (Remarks to the Author):

The authors have addressed all of my comments. Importantly, the experiments are nicely done and they have been quantified and analyzed in robust ways that the field can benefit from for future studies. In addition, the knowledge that they reveal through their work is also interesting and informative for the field of cell biology. In particular, it is nice to see evidence supporting prior models, and building on them in ways that provides more clarity and insight, particularly with the role of anillin in mechanosensing. I also appreciate the authors pointing out that they have laid the foundation for future mechanistic studies where separation-of-function mutations can be used to test some of the finer details of their model.

Notably, many of my major comments reflected issues with the writing rather than with experimental approaches per se. For example, I had struggled to understand some of the methods used, and the authors did a nice job of explaining what they were trying to do and why, and this clarified many of my major issues. While I still may disagree with their choice of language here and there - for example by keeping the word morphogenesis in the first sentence of their abstract, readers could misinterpret what the article is about. While, yes, aberrant cytokinesis could affect the future development of an organism, this isn't the focus of the manuscript as they are not following embryos beyond the first division or assessing fate, etc. However, since this is a personal bias rather than a critical flaw, the authors should be free to describe the major objectives of their article and their findings in their way.

Reviewer #2 (Remarks to the Author):

I greatly appreciate all the additional work the authors put into this manuscript. It is much improved and I am satisfied with the revisions.

Reviewer #3 (Remarks to the Author):

The authors have done an excellent job addressing the criticisms raised in the first round and in my opinion the revised manuscript is suitable for publication

We would like to express our gratitude once again to all the reviewers for providing valuable feedback on our manuscript. We are pleased to see that all the reviewers have now recommended the publication of this paper. In this response, we are addressing the request from the journal to respond to the Reviewer 1's comment (the original comment is highlighted in blue).

Reviewer #1 (Remarks to the Author):

The authors have addressed all of my comments. Importantly, the experiments are nicely done and they have been quantified and analyzed in robust ways that the field can benefit from for future studies. In addition, the knowledge that they reveal through their work is also interesting and informative for the field of cell biology. In particular, it is nice to see evidence supporting prior models, and building on them in ways that provides more clarity and insight, particularly with the role of anillin in mechanosensing. I also appreciate the authors pointing out that they have laid the foundation for future mechanistic studies where separation-of-function mutations can be used to test some of the finer details of their model.

Notably, many of my major comments reflected issues with the writing rather than with experimental approaches per se. For example, I had struggled to understand some of the methods used, and the authors did a nice job of explaining what they were trying to do and why, and this clarified many of my major issues. While I still may disagree with their choice of language here and there - for example by keeping the word morphogenesis in the first sentence of their abstract, readers could misinterpret what the article is about. While, yes, aberrant cytokinesis could affect the future development of an organism, this isn't the focus of the manuscript as they are not following embryos beyond the first division or assessing fate, etc. However, since this is a personal bias rather than a critical flaw, the authors should be free to describe the major objectives of their article and their findings in their way.

Thank you so much for providing valuable feedback to improve our manuscript. Regarding the comment on the abstract, we fully understand the reviewer's perspective. Since the reviewer agrees, we will keep it as it is. The term 'morphogenesis' is also used in the context of studying unicellular organisms (e.g., *Trypanosoma*), and it has room for interpretation in its meanings. As a cell and developmental biologist, I take pride in this personal bias.